# Towards Principled Task Grouping for Multi-Task Learning

**Chenguang Wang**[*]                                   *chenguangwang@link.cuhk.edu.cn*
*School of Data Science*
*The Chinese University of Hong Kong, Shenzhen*

**Xuanhao Pan**[*]                                   *xuanhaopan@link.cuhk.edu.cn*
*School of Data Science*
*The Chinese University of Hong Kong, Shenzhen*

**Tianshu Yu**[†]                                   *yutianshu@cuhk.edu.cn*
*School of Data Science*
*The Chinese University of Hong Kong, Shenzhen*

**Reviewed on OpenReview:** *https://openreview.net/forum?id=3DeSIpzuro*

## Abstract

Multi-task learning (MTL) aims to leverage shared information among tasks to improve learning efficiency and accuracy. However, MTL often struggles to effectively manage positive and negative transfer between tasks, which can hinder performance improvements. Task grouping addresses this challenge by organizing tasks into meaningful clusters, maximizing beneficial transfer while minimizing detrimental interactions. This paper introduces a principled approach to task grouping in MTL, advancing beyond existing methods by addressing key theoretical and practical limitations. Unlike prior studies, our method offers a theoretically grounded approach that does not depend on restrictive assumptions for constructing transfer gains. We also present a flexible mathematical programming formulation that accommodates a wide range of resource constraints, thereby enhancing its versatility. Experimental results across diverse domains, including computer vision datasets, combinatorial optimization benchmarks, and time series tasks, demonstrate the superiority of our method over extensive baselines, thereby validating its effectiveness and general applicability in MTL without sacrificing efficiency. Code is available at `https://github.com/LOGO-CUHKSZ/Principled-Task-Grouping`.

## 1 Introduction

Multitask Learning (MTL) (Caruana, 1997; Zhang & Yang, 2021; Vandenhende et al., 2021) represents a forefront area in machine learning, aiming to improve learning efficiency and prediction accuracy by leveraging commonalities and differences across multiple tasks, reflected by the so-called inter-task "transfer gain". Building upon this foundational concept, MTL has exhibited exceptional performance across a spectrum of domains, including computer vision (Standley et al., 2020; Fifty et al., 2021; Song et al., 2022; Sherif et al., 2023), NLP (Zhang et al., 2022b; Ding et al., 2023), Neural Architecture Search (Guo et al., 2020; Zhang et al., 2022a; Raychaudhuri et al., 2022; Yue et al., 2023), speech recognition (Zhang et al., 2019b; Huang et al., 2022) and combinatorial optimization problems (Wang & Yu, 2023; Wang et al., 2024). Central to the optimization of this framework is the concept of task grouping. Task grouping (Kang et al., 2011; Kumar & Daume III, 2012; Lee et al., 2016; 2018; Zamir et al., 2018; Dwivedi & Roig, 2019; Malhotra et al., 2022; Standley et al., 2020; Fifty et al., 2021; Song et al., 2022) in MTL involves strategically dividing a set of tasks into several groups, where each group encapsulates tasks that share maximal positive transfer while minimizing negative transfer.

---

[*]Equal contribution.
[†]Corresponding author.

Recent studies (Standley et al., 2020; Fifty et al., 2021) have contributed significantly to this domain. The method in Standley et al. (2020) involves training all single-task and two-task networks to build a matrix of transfer gains used to predict the performance of larger groupings. Fifty et al. (2021) adopt a more efficient approach where task affinities are collected during a single run of MTL training. Subsequently, these groups are trained separately using MTL methods. However, these approaches exhibit key limitations. Firstly, there is an absence of a theoretical guarantee in their task affinity measures, raising concerns about the reliability and predictability of the task grouping effectiveness. Secondly, they both rely on an enumeration-based branch and bound algorithm for solving the task grouping problem. This approach not only sacrifices efficiency in terms of computational resources but also poses challenges in incorporating additional constraints, limiting its practical applicability in more complex and realistic scenarios.

In this work, we introduce a novel approach to task grouping in MTL that addresses existing limitations and offers significant advancements over current methodologies. First, we propose a theoretically grounded method for constructing transfer gains. Unlike TAG (Fifty et al., 2021), which assumes restrictive conditions such as convexity and smoothness on loss functions, the proposed transfer gain is derived independently of any conditions. Additionally, it maintains computational complexity at the same order as TAG by adhering to the high-order approximation assumption regarding task relationships, as utilized in prior works (Standley et al., 2020; Fifty et al., 2021). Second, our work introduces a generic and flexible mathematical programming formulation to solve task grouping problems. This formulation can readily incorporate various budget constraints, a critical aspect of real-world applications. By doing so, our method ensures the practicality and adaptability of MTL models in diverse scenarios, ranging from computational budget allocation to resource utilization considerations.

Our experimental evaluations across various domains, including computer vision datasets, combinatorial optimization benchmarks, and time series datasets, demonstrate the validity and generality of our proposed task grouping strategy in three key aspects. First, we establish that our method consistently outperforms a wide range of baselines, encompassing single-task learning, multi-task learning, and various task grouping methods. This substantiates its effectiveness across these three diverse domains. Second, we illustrate the flexibility and effectiveness of our proposed mathematical programming formulation by introducing various constraints, mirroring real-world scenarios where resource budgets, such as GPU memory limitations and resource utilization, come into play. Our results demonstrate that our method significantly outperforms the baseline methods, showcasing its adaptability and performance improvement under such constraints. Finally, we propose two efficiency-enhancing strategies, a sampling approach and a lazy collection mechanism, that substantially reduce the computational overhead of task grouping while maintaining performance quality.

In summary, this work makes several key contributions to the realm of task grouping:

- We propose a theoretically principled approach to constructing transfer gains without relying on restrictive assumptions;

- To solve task grouping problems, we introduce a mathematically generic and flexible programming formulation, capable of seamlessly integrating various budget constraints;

- Through extensive experiments, we demonstrate the effectiveness of our task grouping strategy across diverse domains. Furthermore, we empirically showcase the flexibility of our mathematical programming approach by addressing realistic constraints.

## 2 Related Works

**Task Grouping.** The idea of task grouping is to exploit shared knowledge within each group to improve overall learning efficiency. Early works utilized normalization terms to partition model parameters aligned with task groups (Kang et al., 2011; Kumar & Daume III, 2012; Lee et al., 2016). Lee et al. (2018) extended this approach to deep learning, modeling asymmetric task relationships via autoencoders. Zamir et al. (2018) presented "Taskonomy", which disentangles task relationships based on transfer learning hierarchies. Dwivedi & Roig (2019) introduced representation similarity analysis for task taxonomy, demonstrating effectiveness on the Taskonomy dataset. Malhotra et al. (2022) introduced scheduled task mitigation to dynamically sequence

task learning. Closely related to our work, Standley et al. (2020); Fifty et al. (2021) apply a two-stage methodology: first collecting training information and defining task affinities, then using Branch and Bound algorithms to find optimal groupings. More recent approaches include meta-learning for estimating grouping gains (Song et al., 2022), data-driven methods based on Data Maps (Sherif et al., 2023), differentiable task grouping (Gao et al., 2024), and gradient-based methods for affinity estimation (Li et al., 2024).

**Lookahead Methods.** The philosophy of Lookahead methods is to use future information to guide the current state, which has been widely used in meta-learning (Finn et al., 2017; Nichol et al., 2018; Wang et al., 2020b), multitask learning (Fifty et al., 2021) and optimization techniques (Zhang et al., 2019a; Wang et al., 2020a; Zhou et al., 2021; Byun et al., 2022). In particular for multitask learning, Fifty et al. (2021) collected the one-step-forward loss information between task pairs for each gradient updating and constructed the overall task affinity matrix at the end of training.

**Loss Balance.** Numerous works have emerged to address multitask learning by exploring the balance among the losses from different tasks (Mao et al., 2021; Yu et al., 2020; Javaloy & Valera, 2022; Navon et al., 2022; Kendall et al., 2018; Liu et al., 2021a;b; Guangyuan et al., 2022; Liu et al., 2022). In these works, various loss reweighing mechanisms are designed to dynamically balance the importance of each task, which can relieve the negative transfer among tasks in terms of gradient information.

## 3 Preliminary

We first establish the formal definitions and notations that form the foundation of our approach.

**Definition 1** (Multitask Learning). *Consider a set of tasks $\mathcal{T} = \{T_i \mid i \in [n]\}$, where $[n] = \{1, 2, \ldots, n\}$ and each task $T_i$ is associated with a learning objective $L_i(\phi, \theta_i)$. Here, $\phi \in \mathbb{R}^p$ represents the shared parameters across all tasks, and $\theta_i \in \mathbb{R}^{p_i}$ denotes the task-specific parameters for task $T_i$. The multitask learning objective is to jointly minimize a weighted combination of individual task losses:*

$$\phi^*, \{\theta_i^*\}_{i=1}^n = \arg \min_{\phi, \{\theta_i\}_{i=1}^n} \sum_{i=1}^n \lambda_i L_i(\phi, \theta_i), \tag{1}$$

*where $\{\lambda_i > 0\}_{i=1}^n$ are task-specific weights that balance the contribution of each task.*

While conventional multitask learning directly optimizes the shared and task-specific parameters through the combined objective, our approach exploits the inherent relationships between tasks during the training process. Rather than training all tasks jointly within a single optimization framework, we organize them into groups based on their mutual influences, leading to the concept of task grouping.

**Definition 2.** *(Task Grouping) Let $\mathcal{T} = \{T_i \mid i \in [n]\}$ denote the set of tasks, and $G = \{G_j \mid j \in [m]\}$ represent the set of task groups. Task grouping aims to establish a mapping based on task relationships such that for every task $T_i$, there exists a group $G_j$ to which $T_i$ is assigned, ensuring the inclusion of at least one task in each group and resulting in the best performance for each task.*

After establishing the task groups, we optimize parameters separately within each group:

$$\min_{\phi_j, \{\theta_i\}_{i \in G_j}} \sum_{i \in G_j} \lambda_i L_i(\phi_j, \theta_i), \ \forall G_j \in G \tag{2}$$

where $\phi_j$ represents the shared parameters specific to group $G_j$. This group-wise optimization approach enhances both computational efficiency and learning effectiveness by allowing each group to focus on a coherent set of related tasks.

**Relationship between Equations equation 1 and equation 2.** Equation equation 1 is the standard MTL objective in which all tasks share a single global parameter $\phi$. A key challenge is that jointly optimizing all tasks under Equation equation 1 often causes negative transfer: gradients from unrelated tasks interfere with one another, degrading per-task performance. Task grouping addresses this by partitioning tasks into groups so that within each group tasks exhibit strong mutual positive transfer, while harmful cross-group interactions are eliminated by assigning each group its own dedicated shared parameter $\phi_j$. Importantly,

Equation equation 2 is a *strict generalization* of Equation equation 1: when $m = 1$ (all tasks in one group), $\phi_1 = \phi$ and Equation equation 2 reduces exactly to Equation equation 1. Task grouping therefore strictly expands the space of training configurations beyond standard MTL. When task relationships are heterogeneous, a well-chosen grouping under Equation equation 2 can alleviate the negative transfer that would arise under Equation equation 1, yielding improved per-task performance — as consistently demonstrated in Section 5.

## 4 Method

To infer task groupings for subsequent optimization processes in Equation equation 2, we introduce a methodology for constructing transfer gains, as elucidated in Section 4.1, demonstrating its efficacy in yielding theoretical outcomes without relying on extra assumptions. Subsequently, we propose a versatile mathematical programming framework in Section 4.2 that flexibly accommodates various budget constraints. This formulation is instrumental in deriving the outcomes of task grouping. Furthermore, in Section 4.3, we conduct a detailed analysis of the computational complexity associated with collecting transfer gains, in comparison to TAG (Fifty et al., 2021).

### 4.1 Assumption-Free Transfer Gain

In this subsection, we introduce the pivotal concept of proposed transfer gain in our method.

**Definition 3.** *(Transfer Gain) For tasks $T_i \neq T_j$, the* task transfer gain *from $T_i$ to $T_j$ at training step $t$ is characterized by:*

$$\mathcal{S}_{i \to j}^t = 1 - \frac{L_j\left(\phi_{\{i,j\}}^{t+1}, \theta_j^{t+1}\right)}{L_j\left(\phi_{\{j\}}^{t+1}, \theta_j^{t+1}\right)}. \tag{3}$$

*In this equation, $L_j$ represents a task-specific metric, such as the objective to be minimized (see Remark 1 for domain-specific instantiations). Note that $\theta_j^{t+1}$ in both the numerator and denominator refers to the* same *fixed quantity — the task-specific parameters of $T_j$ updated from $T_j$'s own gradient. The distinction lies solely in the shared parameter $\phi$: the numerator uses $\phi_{\{i,j\}}^{t+1}$, obtained by updating $\phi^t$ with the* joint *gradient of $T_i$ and $T_j$, whereas the denominator uses $\phi_{\{j\}}^{t+1}$, obtained using only $T_j$'s gradient. We then define the* group transfer gain *from any $A \subseteq \mathcal{T}$ (with $j \notin A$) to task $T_j$ as:*

$$\mathcal{S}_{A \to j}^t = 1 - \frac{L_j\left(\phi_{A \cup \{j\}}^{t+1}, \theta_j^{t+1}\right)}{L_j\left(\phi_{\{j\}}^{t+1}, \theta_j^{t+1}\right)}. \tag{4}$$

*Furthermore, we extend this concept to define the* group transfer gain *from any $A \subseteq \mathcal{T}$ to $B \subseteq \mathcal{T}$ as:*

$$\mathcal{S}_{A \to B}^t = \sum_{j \in B} \mathcal{S}_{A \to j}^t, \tag{5}$$

*which allows us to measure the collective transfer of knowledge between groups of tasks.*

*Remark* 1 (Instantiation of $L_j$ across Domains). The metric $L_j$ in Definition 3 is instantiated according to the task type in each experimental domain: **Taskonomy** (depth, surface normal, keypoint, edge, segmentation): task-specific regression training loss; **CelebA** (facial attribute classification): binary cross-entropy training loss; **COP** (TSP, CVRP, OP): the objective value function within the POMO framework (Kwon et al., 2020); **ETTm1** (time-series forecasting): mean absolute error (MAE) training loss. In all cases $L_j$ is evaluated on training data at each step $t$, so that transfer gains reflect in-training dynamics rather than held-out generalization.

While our formulation, as presented in Equation equation 3, bears a formal resemblance to TAG (Fifty et al., 2021), there are essential differences between the two. We will elucidate these distinctions and demonstrate the superior advantages of our approach in the ensuing discussion.

**First**, Fifty et al. (2021) defines task affinity as $\mathcal{Z}_{i \to j}^t = 1 - \frac{L_j\left(\phi_{\{i\}}^{t+1}, \theta_j^t\right)}{L_j\left(\phi^t, \theta_j^t\right)}$ with respect to the loss function, reflecting the effects of training $T_i$ on $T_j$, while $\mathcal{S}_{i \to j}^t$ measures the effects of training $T_i$ on *training $T_j$*. This substantial distinction allows us to establish the relationship between loss decrease and the value of $\mathcal{S}_{i \to j}^t$ without additional assumptions. Specifically, if $\mathcal{S}_{i \to j}^t > \mathcal{S}_{k \to j}^t$, then training $\{T_i, T_j\}$ results in a greater loss decrease than training $\{T_k, T_j\}$, as summarized in the following observation:

**Observation 1.** *If $\mathcal{S}_{A_1 \to i}^t > \mathcal{S}_{A_2 \to i}^t$, then training task group $A_1$ induces a larger loss decrease than $A_2$ for task $i$.*

In contrast, Fifty et al. (2021) introduces restrictive constraints, such as strong convexity on loss functions, to enforce this relationship. **Second**, in Fifty et al. (2021), group transfer gain is formulated as $\mathcal{Z}_{\{j,k\} \to i}^t = \frac{1}{2}\left(\mathcal{Z}_{j \to i}^t + \mathcal{Z}_{k \to i}^t\right)$. This formulation directly defines the group transfer gain at the task level, lacking theoretical guarantees of effectiveness. In this work, we establish the connection between task and group transfer gain in Proposition 1, providing both theoretical advantages and valid empirical operations at the implementation level.

Table 1 summarises the three key dimensions along which $\mathcal{S}_{i \to j}^t$ differs from TAG's affinity $\mathcal{Z}_{i \to j}^t$. With respect to **measurement target**, TAG measures the effect of applying $T_i$'s gradient to the *frozen* parameters $(\phi^t, \theta_j^t)$ of $T_j$, computing a counterfactual quantity that does not reflect actual co-training dynamics. Our $\mathcal{S}_{i \to j}^t$ instead measures the effect of *jointly* training $\{T_i, T_j\}$ on $T_j$'s training outcome with both shared and task-specific parameters updated, directly quantifying the benefit of co-training in the MTL sense. Regarding **group-level theoretical guarantees**, TAG's arithmetic-average definition of group affinity carries no formal justification, while Proposition 1 provides an explicit error bound whose magnitude is controlled by the learning rate, group size, and Lipschitz constant — all small under standard training conditions. Finally, in terms of **empirical robustness**, TAG relies on convexity and smoothness assumptions that are unlikely to hold for combinatorial optimization and time-series losses; this explains why TAG performs comparably to or worse than random grouping on COP and ETTm1 (Figure 1, Tables 7–8), whereas our method remains consistently superior across all four domains.

Table 1: Comparison between TAG's affinity $\mathcal{Z}_{i \to j}^t$ and our transfer gain $\mathcal{S}_{i \to j}^t$.

| **Dimension** | **TAG** ($\mathcal{Z}_{i \to j}^t$) | **Ours** ($\mathcal{S}_{i \to j}^t$) |
|---|---|---|
| Measurement target | Effect of $T_i$ alone on frozen $(\phi^t, \theta_j^t)$ | Effect of jointly training $\{T_i, T_j\}$ on $T_j$ |
| Group-level guarantee | Arithmetic average (no justification) | Proposition 1: explicit error bound |
| Required assumptions | Strong convexity & smoothness of losses | None (Observation 1 holds unconditionally) |
| Empirical robustness | Degrades on COP & ETTm1 | Consistent across all four domains |

**Proposition 1.** *Consider a multi-task learning setup with shared parameters $\phi \in \mathbb{R}^d$ and task-specific parameters $\theta_k$ for each task $T_k \in \mathcal{T}$. Let $L_k(\phi, \theta_k)$ be the loss function for task $T_k$. Suppose the model parameters are updated from $(\phi^t, \{\theta_k^t\}_{k \in \mathcal{T}})$ to $(\phi^{t+1}, \{\theta_k^{t+1}\}_{k \in \mathcal{T}})$ using a single step of gradient descent with learning rate $\eta_t > 0$. Assume the following conditions hold:*

1. *For all tasks $k \in \mathcal{T}$, the loss function $L_k(\phi, \theta_k)$ is l-Lipschitz with respect to $\phi$ for any fixed $\theta_k$, and differentiable with respect to $\phi$. This implies that $||\nabla_\phi L_k(\phi, \theta_k)|| \le l$ for all $\phi, \theta_k$.*

2. *There exists a constant $C > 0$ such that $L_j\left(\phi_{\{j\}}^{t+1}, \theta_j^{t+1}\right) \ge C$. This condition is required to hold only for the single-task trained parameters $\phi_{\{j\}}^{t+1}$ (when $T_j$ is trained in isolation), as this term appears exclusively as the denominator in $\mathcal{S}_{i \to j}^t$ and $\mathcal{S}_{A \to j}^t$. It places no constraint on loss values under joint training. Standard regularization (e.g., weight decay, dropout) and early stopping naturally prevent the single-task loss from reaching zero in practice.*

*Let $\mathcal{S}_{i \to j}^t$ and $\mathcal{S}_{A \to j}^t$ be the task transfer gain and group transfer gain as defined in Equation 3 and 4. Then, we have the following bound on the absolute difference between the group transfer gain and the average of*

*individual task transfer gains:*

$$\left| \mathcal{S}_{A \to j}^t - \frac{1}{|A|} \sum_{i \in A} \mathcal{S}_{i \to j}^t \right| \leq \frac{\eta_t (1 + |A|) l^2}{C}. \tag{6}$$

*Remark* 2 (Practical Implications of the Bound's Magnitude). In practical training scenarios, several factors contribute to making this upper bound small, supporting the notion that the group transfer gain is often well-approximated by the average of individual transfer gains: **Learning Rate:** Standard optimization practices for neural networks employ relatively small learning rates, and often use schedules that decrease $\eta_t$ over time. A small $\eta_t$ directly scales down the entire bound; **Group Size:** The number of source tasks included in a group for simultaneous training is typically constrained by available computational resources, such as GPU memory and processing power; **Loss Lower Bound:** The condition $L_j(\phi_{\{j\}}^{t+1}, \theta_j^{t+1}) \geq C > 0$ is often met or promoted through standard training techniques. Regularization methods (e.g., L2 regularization, dropout) and early stopping prevent the loss function from converging to zero; **Lipschitz Constant:** While the Lipschitz constant $l$ depends on the choice of model architecture and loss function, for many commonly used models and loss functions (e.g., truncated gradients for bounded inputs), $l$ is finite.

Collectively, the use of small learning rates, practical limits on group size, and techniques that ensure a positive lower bound on the loss contribute to making the upper bound $\frac{\eta_t(1+|A|)l^2}{C}$ small.

Based on these findings, $\mathcal{S}_{i \to j}^t$ serves as a measure of intrinsic signals that elucidate the relationships between tasks during training. Furthermore, we introduce the concepts of cumulative transfer gain and group transfer throughout the training process. These are defined as follows:

$$\mathcal{S}_{i \to j} = \frac{1}{T} \sum_{t=1}^{T} \mathcal{S}_{i \to j}^t, \qquad \mathcal{S}_{A \to j} = \frac{1}{|A|} \sum_{i \in A} \mathcal{S}_{i \to j}. \tag{7}$$

These metrics quantify the impact of training group $A$ on task $j$ over the entire training period.

## 4.2 Generic and Flexible Task Grouping Framework

After obtaining the cumulative transfer gain $\mathcal{S}$ from Equation equation 7, the remaining challenge is to accurately determine the task grouping results. Practical applications often require customization based on various constraints, such as resource limitations. Therefore, it is crucial to develop a general and flexible task grouping framework. Unlike previous approaches (Zamir et al., 2018; Fifty et al., 2021; Standley et al., 2020), which utilize Binary Integer Programming and Branch & Bound methods, we propose a mathematical programming framework that can be transformed into Mixed Integer Programming. This involves setting binary variables $X_{ij}$, where $i \in [n]$ and $j \in [m]$, with $X_{ij} = 1$ indicating the assignment of task $i$ to group $j$, and $X_{ij} = 0$ otherwise. Notably, $X_{\cdot j} \in \mathbb{R}^n$ represents the $j$-th column of $X$. The vector $\mathbf{1}$ is composed entirely of ones with an adaptable dimension. The element $B_{ij}$ in $B \in \mathbb{R}^{n \times m}$ denotes the budget associated with task $T_i$ assigned to group $G_j$. The vector $\mathbf{b} \in \mathbb{R}^m$ represents the maximum budget for each group. The symbol "$\odot$" signifies the element-wise product between matrices, resulting in the following formulation:

$$
\begin{aligned}
\max_X \quad & \sum_{j=1}^{m} \frac{1}{\mathbf{1}^\top X_{\cdot j}} X_{\cdot j}^\top \mathcal{S} X_{\cdot j} \\
\text{s.t.} \quad & X^\top \mathbf{1} \geq \mathbf{1}, \quad X\mathbf{1} \geq \mathbf{1}, \quad (B \odot X)^\top \mathbf{1} \leq \mathbf{b} \\
& \|X_{\cdot j_1} - X_{\cdot j_2}\|^2 \geq 1 \text{ for } j_1 \neq j_2, \quad X \in \{0,1\}^{n \times m}.
\end{aligned}
\tag{8}
$$

The primary objective of the task grouping problem is to compute the aggregate impact of training each group individually. The quadratic form $X_{\cdot j}^\top \mathcal{S} X_{\cdot j}$ arises from the approximation stated in Proposition 1. Several key constraints are imposed to ensure a meaningful solution. The constraint $X^\top \mathbf{1} \geq \mathbf{1}$ is established to guarantee that each group contains at least one task, while the constraint $X\mathbf{1} \geq \mathbf{1}$ ensures that all tasks are incorporated into the grouping outcomes. Additionally, diverse budgetary constraints can be introduced by incorporating the inequality $(B \odot X)^\top \mathbf{1} \leq \mathbf{b}$.

Moreover, it is imperative that the resulting groups remain distinct, and this distinctiveness is ensured by the condition $\|X_{\cdot j_1} - X_{\cdot j_2}\|^2 \geq 1$. Remarkably, recent advancements in powerful mathematical programming solvers have significantly enhanced the efficiency of resolving these problems through appropriate transformations. We present the transformation of Formulation 8 into a Mixed-Integer Quadratic Programming (MIQP) problem with non-linear constraints. This involves introducing a continuous variable $\mathbf{y} \in [0, 1]^n$ and binary variables $Z_{ijk}$ to obtain:

$$
\begin{aligned}
\max_{X,y} \quad & \sum_{j=1}^{m} \sum_{k=1}^{n} \sum_{i=1}^{n} S_{ik} y_j Z_{ijk} \\
\text{s.t.} \quad & X^T \mathbf{1} \geq \mathbf{1}, \quad X\mathbf{1} \geq \mathbf{1} \\
& (B \odot X)^T \mathbf{1} \leq b, \quad (X^T \mathbf{1}) \odot \mathbf{y} = \mathbf{1} \\
& Z_{ijk} = X_{ij} \cdot X_{kj} \quad \forall i, j, k \\
& \|X_{\cdot j_1} - X_{\cdot j_2}\|^2 \geq 1, \quad j_1 \neq j_2 \\
& X \in \{0, 1\}^{n \times m}
\end{aligned}
\tag{9}
$$

which can be solved using classical solvers; in this work, we employ Gurobi (Gurobi Optimization, LLC, 2023). Detailed results of the comparison between different solvers are provided in Section 5.5.

### 4.3 Analysis of Computation Complexity

In this section, we discuss the computational complexity of the proposed method for collecting transfer gains and compare it to the most closely related work, TAG (Fifty et al., 2021). We begin by defining the relevant notations: For each task $T_i$, the computational costs of a feed-forward pass and a backward pass are denoted as $\mathcal{F}_i$ and $\mathcal{B}_i$, respectively. We introduce $\mathcal{C}_i$ to represent the computational cost of parameter operations (e.g., weight assignment), which is proportional to the number of model parameters. The average costs for these tasks are represented by $\mathcal{F} = \frac{1}{n} \sum_{i=1}^{n} \mathcal{F}_i$, $\mathcal{B} = \frac{1}{n} \sum_{i=1}^{n} \mathcal{B}_i$, and $\mathcal{C} = \frac{1}{n} \sum_{i=1}^{n} \mathcal{C}_i$. Based on these notations, the computation costs for TAG and our proposed method are $(n^2 + n)\mathcal{F} + n\mathcal{B} + n\mathcal{C}$ and $(n^2 + 2n)\mathcal{F} + n\mathcal{B} + (n^2 + n)\mathcal{C}$, respectively. Both methods have the same order of computational complexity with respect to the number of tasks. While our method involves an additional $n$ feed-forward computations and incurs a higher cost for parameter operations, it is important to emphasize that the costs of feed-forward and parameter updates are typically much smaller than the backward pass cost ($\mathcal{B}$) in practice. Consequently, the impact of these additional computations is minimal, especially when considering the theoretical advantages our method offers.

To further enhance computational efficiency when the number of tasks is large, we propose two practical approaches for collecting transfer gains: (1) **Sampling strategy**: We implement a sampling-based approach for collecting transfer gains. We define a random variable $T$ that follows a uniform distribution, denoted as $T \sim \text{Unif}\{1, 2, \ldots, n\}$. A subset of tasks of size $T$ is then randomly selected, and transfer gains are gathered solely from this subset. This sampling strategy significantly reduces the computational cost of our method to $\frac{n^2 + 6n + 2}{3}\mathcal{F} + n\mathcal{B} + \frac{(n+1)(n+5)}{6}\mathcal{C}$ in expectation, which is substantially lower than that of TAG. (2) **Lazy collection strategy**: Employing a lazy collection strategy for transfer gains maintains performance while reducing computational costs. The analysis reveals that cumulative transfer gain varies across different training phases, yielding diverse outcomes. These insights allow for substantial reductions in computational costs. Detailed analysis and results are provided in Section 5.4.

## 5 Experiments

In this section, we present the experimental evaluation of our task grouping approach, demonstrating its effectiveness on computer vision, combinatorial optimization, and time series analysis. Due to space constraints, only the primary experimental settings and results are included. See comprehensive details about experimental setups, analysis of group results, and digital tables in Appendix B.

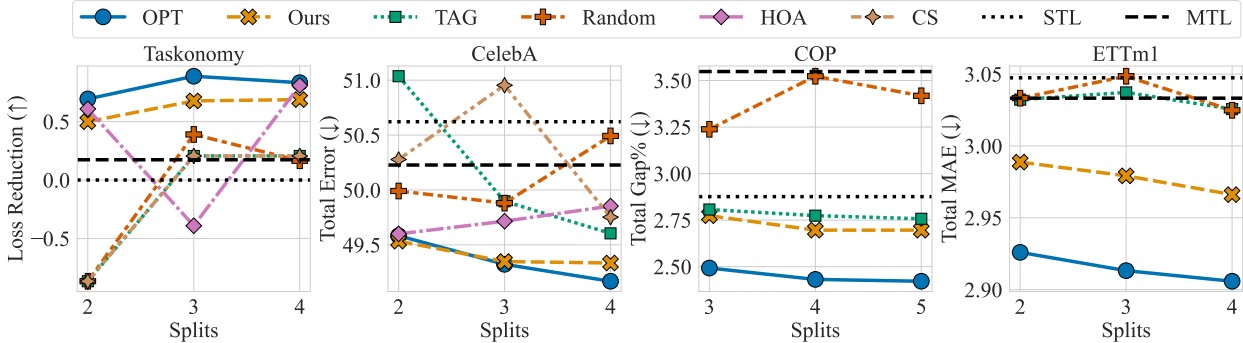

Figure 1: Performance demonstration across grouping methods on each dataset. This figure presents the loss reduction (↑) for Taskonomy, total test error (↓) for CelebA, total optimality gap (↓) for COP and total MAE (↓) for ETTm1, segmented by various data splits. MTL denotes joint training of all tasks in one group. Full results including all baselines are provided in Tables 5-8. All results are averaged over 3 runs with different random seeds.

## 5.1 Experimental Setups

**Datasets and Metrics.** Our experiments are designed to validate the versatility and superiority of our method across four datasets: (1) Taskonomy (Zamir et al., 2018) and (2) CelebA: These are classical computer vision datasets used in previous task grouping methods. Following the settings in TAG (Fifty et al., 2021), we selected five tasks from Taskonomy and nine tasks from CelebA for our experiments. The performance of each method is evaluated by the total loss reduction compared with single-task learning (STL) for Taskonomy and the classification error rates for CelebA across all tasks. To ensure experimental consistency, we adhere to the network architecture and training hyperparameters specified in TAG (Fifty et al., 2021); (3) Combinatorial Optimization Benchmarks (Wang & Yu, 2023): We test six tasks: TSP20, TSP50, CVRP20, CVRP50, OP20, and OP50, representing various scales of the Traveling Salesman Problem (TSP), Capacitated Vehicle Routing Problem (CVRP), and Orienteering Problem (OP). Performance is evaluated based on the average optimality gap, mathematically defined as:

$$\text{Gap}\% = \frac{1}{N} \sum_{i=1}^{N} \left( 1 - \frac{\text{solver}(\mathcal{I}_i)}{\text{gt}(\mathcal{I}_i)} \right) \times 100,$$

evaluated over $N = 10,000$ instances for each task to measure the solution's deviation from the ground truth obtained from Gurobi (Gurobi Optimization, LLC, 2023). The neural solver used for these tasks is the POMO framework (Kwon et al., 2020), noted for its effectiveness in addressing combinatorial optimization problems; (4) ETTm1 (Wu et al., 2021): This is an electric load dataset with seven time series. Effectiveness is assessed using the mean absolute error's (MAE) relative reduction as the evaluation metric. We employ the model based on the AutoFormer framework (Wu et al., 2021), specifically designed for the intricacies and predictive challenges of multivariate time series data.

**Baselines.** Our experimental evaluation involves a comprehensive comparison against a range of established methods: (1) Single Task Learning (STL); (2) MTL methods: We consider a variety of MTL methods that employ different strategies for joint task learning, including: Naive-MTL, Bandit-MTL (Mao et al., 2021), PCGrad (Yu et al., 2020), Nash-MTL (Navon et al., 2022), Uncertainty-Weighting (UW) (Kendall et al., 2018), LinearScale, GradNorm, CAGrad, and Linear Surrogate (Li et al., 2023); (3) Task grouping methods: Random policy by which tasks are grouped randomly and results are taken the average for 10 repeats; Optimal policy which is obtained by enumeration; TAG (Fifty et al., 2021), a known state-of-the-art (SOTA) task grouping method to group tasks based on their affinity. Additionally, we include high order approximation (HOA) (Standley et al., 2020), cosine similarity (CS), MTG (Song et al., 2022), STG (Jeong & Yoon, 2025), and Grad-TAG (Li et al., 2024). Comprehensive results for these additional baselines are provided in the Appendix B.

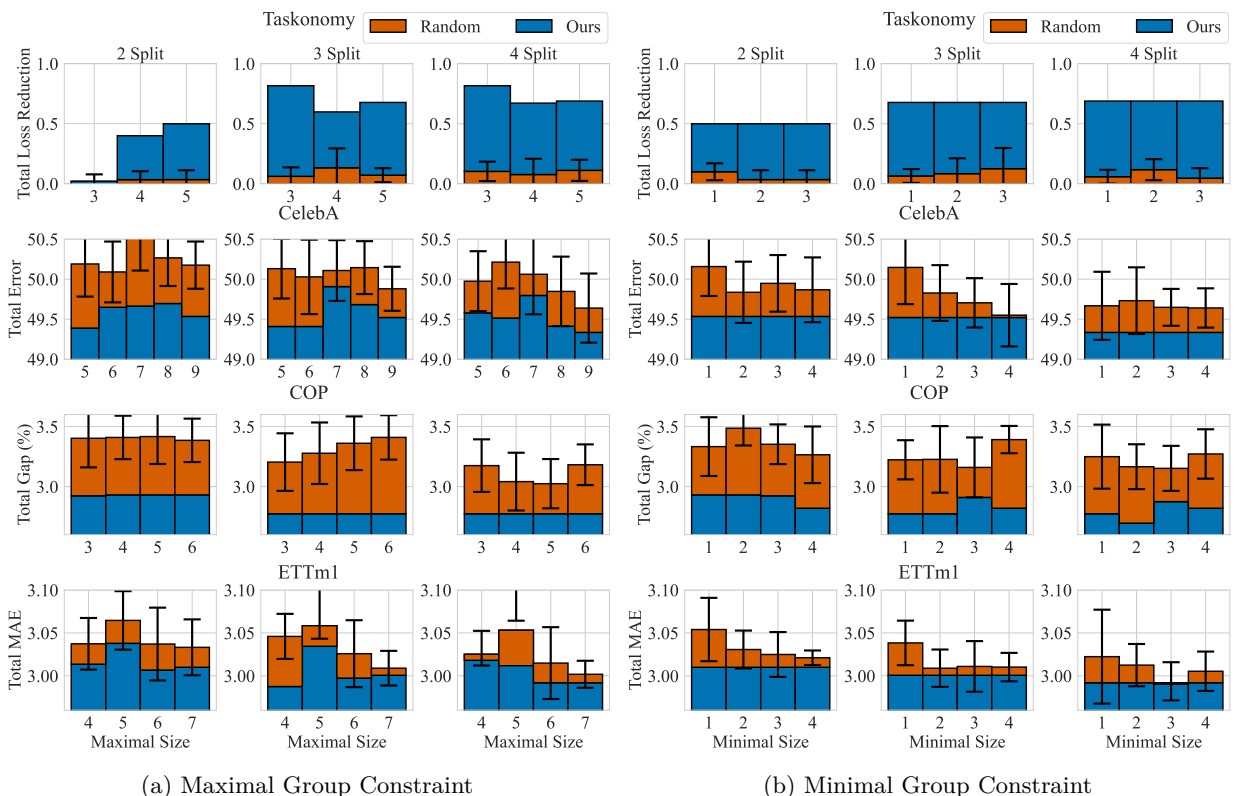

(a) Maximal Group Constraint          (b) Minimal Group Constraint

Figure 2: Comparative Performance under Maximum and Minimum Group Size Constraints. The figure delineates the performance of our task grouping method against random policy. Metrics such as loss reduction (↑) for Taskonomy dataset, total error (↓) for the CelebA dataset, Total Gap (↓) for combinatorial optimization problems (COP), and Total Mean Absolute Error (MAE) (↓) for time series forecasting tasks are evaluated across a range of group sizes, illustrating the adaptability of our method to both maximal and minimal size constraints.

## 5.2 Main Results

In this subsection, we present the empirical evaluation of our task grouping approach across four diverse domains: computer vision (Taskonomy and CelebA datasets), combinatorial optimization problems (COP), and time series forecasting (ETTm1 dataset). For brevity, we present the most significant findings from Figure 1 here, while comprehensive digital tables with detailed results are provided in the Appendix: Taskonomy results in Section B.3, CelebA results in Section B.4, combinatorial optimization results in Section B.5, and time series forecasting results in Section B.6.

**Results on Taskonomy.** In Figure 1, the leftmost figure for the Taskonomy dataset focuses on evaluating the loss reduction compared to STL, where higher values indicate better performance. Results show that our method consistently outperforms the STL and MTL benchmarks across all splits, demonstrating robust and reliable performance compared to traditional learning approaches. Furthermore, our method shows a clear trend of improving performance with an increasing number of splits, indicating that finer granularity in task grouping leads to better results. While HOA performs well in splits 2 and 4, it shows a significant drop in performance at split 3, highlighting its lack of stability. Although our method occasionally performs slightly worse than HOA in certain splits, it provides superior overall stability and consistency. Given that HOA's high computational cost makes it less practical, the proposed method stands out as the best choice for achieving a balance between performance, stability, and computational efficiency. Further digital results and analysis are in Appendix B.3.

**Results on CelebA.** The second-left figure in Figure 1 provides a comprehensive evaluation of various baselines in terms of the total error metric for the CelebA dataset, where lower values indicate better performance. It is observed that MTL exhibits superior performance compared to STL, and most grouping methods result in a further reduction in total error. This suggests the presence of complex underlying relationships between the tasks in this dataset, indicating that our grouping methods can identify specific combinations that enhance overall performance. Our method consistently outperforms all baselines in terms of total error across all splits, with the only exception being the optimal grouping strategy found in TAG Fifty et al. (2021). Additionally, our method effectively leverages the granularity afforded by increased splits, demonstrating a decrease in total error. Detailed results are elaborated in Appendix B.4.

**Results on COP** In the comparative analysis presented in Figure 1, STL demonstrates a robust baseline, outperforming MTL methods with respect to the total gap metric. Within the domain of task grouping methods, both TAG and our proposed method have shown the capability to surpass the STL baseline in certain aspects. Notably, our method consistently achieves the best performance among non-optimal baselines across each grouping strategy, indicating its efficacy in handling multiple related tasks simultaneously. As we consider the performance trends across different task groupings, it is observed that the efficacy of Optimal, TAG, and our method improves as the splits become larger. This trend suggests that the tasks within the COP benchmark exhibit high positive transfer potential. Detailed grouping analysis can be seen in Appendix B.5, where our method achieves logical groupings of tasks, pairing the same types of COPs together: (TSP20, TSP50), (CVRP20, CVRP50), and (OP20, OP50) when splitting these tasks into three groups. This reflects an intuitive understanding that tasks of the same types benefit from being trained together.

**Results on ETTm1.** The rightmost figure in Figure 1 evaluates the total MAE for the ETTm1 dataset, where lower values indicate better performance. It shows that our method consistently outperforms all non-optimal benchmarks across all splits. Moreover, it demonstrates a trend of improving performance with an increasing number of splits, highlighting its robustness and reliability in minimizing MAE. However, TAG shows performance comparable to the Random baseline in most splits and exceeds both STL and MTL benchmarks only at the 4-split level. This variability in TAG's performance suggests that it struggles to identify and leverage the intricate connections between time series tasks. In contrast, the superior performance of our method highlights its capability to effectively uncover and utilize the intrinsic relationships between time series tasks, leading to the best overall results. See detailed results in Appendix B.6.

**Overall Analysis.** In summary, our proposed method demonstrates superior robustness across four distinct domains, ranging from semantic visual features (Taskonomy, CelebA) to the mathematical constraints of optimization (COP) and stochastic temporal dynamics (ETTm1). A key finding is that while baselines like TAG or HOA perform adequately in vision, they often falter in COP or Time Series. Our method's consistent dominance across this diverse spectrum validates our theoretical premise: by constructing transfer gains without restrictive assumptions, we achieve a grouping strategy that is not merely effective for standard visual tasks but genuinely robust to the varying loss landscapes of broader machine learning applications.

### 5.3 Task Grouping with Constraints

In practical scenarios, constraints such as limited computational resources, data availability, and group size requirements are common. Tasks often involve varying data acquisition costs, necessitating budget management. In distributed learning, where resources are distributed across nodes, complying with group size limits is crucial. Our mathematical programming approach, as outlined in Formulation 8, effectively addresses the task grouping challenge under these constraints by incorporating customized limitations to meet the practical demands of real-world applications. In this section, we address the constraint that group sizes must fall within a specified range, represented as: $\mathbf{m_1} \leq X^T \mathbf{1} \leq \mathbf{m_2}$, where $\mathbf{m_1}, \mathbf{m_2} \in \mathbb{R}^m$, using element-wise comparison. Our experiments in computer vision, COP, and time series applications demonstrate the model's ability to adhere to these size constraints. We benchmark our method against random sampling, conducted ten times under the same constraints, with the experimental setup detailed subsequently.

**Maximum Group Size Constraint.** For this constraint, we define $\mathbf{m_1}$ as $\mathbf{1}$ and $\mathbf{m_2}$ as $M\mathbf{1}$, where $M$ represents the maximum permissible group size, dictated by memory limitations. For the Taskonomy dataset containing five tasks, we select $M$ from $\{3, 4, 5\}$. In the context of the CelebA dataset, which comprises

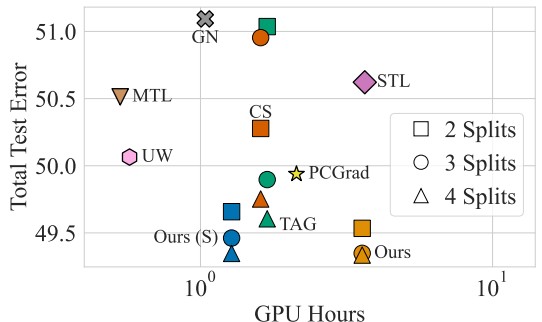

| Freq. | Relative Improvement (↑) | | Relative Speedup (↑) | |
|---|---|---|---|---|
| | CelebA | ETTm1 | CelebA | ETTm1 |
| 1 | 3.50% | 5.76% | 1.00 | 1.00 |
| 5 | 3.53% | 9.48% | 4.25 | 4.62 |
| 10 | 3.94% | 9.20% | 7.18 | 8.45 |
| 25 | 4.60% | 8.75% | 12.16 | 16.64 |
| 50 | 5.13% | 8.75% | 15.95 | 24.94 |
| 100 | 1.83% | 6.70% | 15.95 | 33.24 |
| 200 | 3.06% | 5.87% | 18.62 | 39.16 |

Figure 3: Average classification error for 2, 3, and 4-split task groupings for the subset of 9 tasks in CelebA, compared across various methods (Ours (S), Ours, TAG, CS, STL, MTL, UW, GN, PCGrad) versus GPU hours.

Table 2: Test errors across different frequencies of transfer gain collection, compared to test errors from random groupings, with speedup evaluated based on the computation of transfer gains at each step.

nine tasks, we set $M$ within the range $\{5, 6, 7, 8, 9\}$. For the COP benchmark involving six tasks, $M$ is selected from $\{3, 4, 5, 6\}$. For time series tasks comprising seven tasks, we choose $M$ from $\{4, 5, 6, 7\}$. The experimental results, as illustrated in Figure 2a, exhibit a uniform trend across varying cases and group sizes. For example, in the Taskonomy and CelebA datasets, our method consistently exhibits superior performance compared to the random policy. This is evident in the progressively larger loss reduction and lower total error rates as the maximum group size increases. Similarly, in the COP benchmarks and time series tasks with their respective maximum group size constraints, our method maintains or even enhances its performance. This consistent trend, observed across various maximum group sizes and tasks, underscores the robustness and adaptability of our approach in accommodating changing task constraints.

**Minimum Group Size Constraint.** In this case, the vectors $\mathbf{m_1}$ and $\mathbf{m_2}$ are defined as $m\mathbf{1}$ and $M_{\max}\mathbf{1}$, respectively, where $M_{\max}$ denotes the total number of tasks specific to each case: five for the Taskonomy dataset, nine for the CelebA dataset, six for COP, and seven for time series tasks. This formulation ensures that device utilization at each node surpasses a certain threshold, thereby guaranteeing efficient usage. For the Taskonomy dataset, we select $m$ from $\{1, 2, 3\}$. For the remaining datasets, $m$ is chosen from the set $\{1, 2, 3, 4\}$. Results are presented in Figure 2b. In the CV datasets, as the minimum group size increases, our method consistently demonstrates superior performance compared to the random policy, indicating improved efficiency in handling larger group sizes. Similarly, in the COP benchmarks, the Total Gap percentage decreases as the group size grows, highlighting our method's effectiveness under tighter constraints. Although the performance of our method in the time series tasks under minimum group size constraints does not match the levels achieved in the CV and COP benchmarks, it still significantly outperforms the random policy. This performance stability highlights the efficacy of our method in the presence of minimal size constraints.

### 5.4 Improve Training Efficiency for Task Grouping

As discussed in Section 4.3, computational complexity becomes a significant concern when applying our task grouping method to scenarios with a large number of tasks. Here, we empirically evaluate the two efficiency-enhancing strategies proposed earlier: the sampling strategy and the lazy collection strategy. Our experiments demonstrate that these approaches substantially reduce computational overhead while maintaining the quality of task groupings.

**Sampling Strategy.** We evaluate our sampling strategy on the CelebA dataset (9 tasks) by comparing four grouping methods in Figure 3: our original approach, TAG, CS, and our sampling-enhanced method (Ours (S)). We also benchmark against five non-grouping methods: STL, MTL, UW, GradNorm, and PCGrad. The analysis focuses on both computational efficiency and performance effectiveness. Our original method achieves the lowest total test error across all splits, demonstrating superior performance but at the cost of increased runtime. By incorporating the sampling strategy, Ours (S) significantly reduces computational

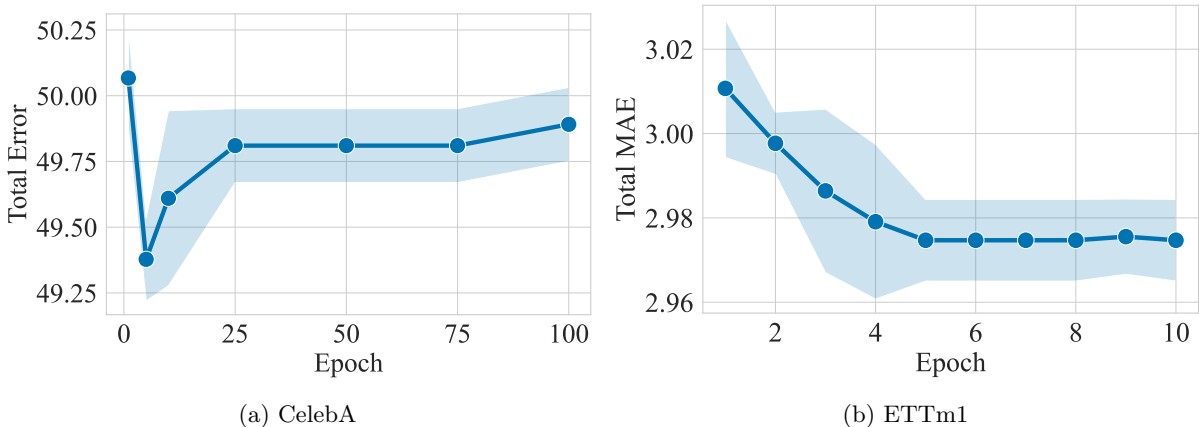

(a) CelebA  (b) ETTm1

Figure 4: This graph illustrates the performance trends of groups generated by our method, as a function of the number of epochs involved in the construction of transfer gains. Each point represents the mean total error at a specific epoch.

overhead while maintaining comparable performance. Notably, Ours (S) outperforms TAG in both accuracy and efficiency, validating its effectiveness.

**Wall-clock Time Comparison with TAG.** Table 3 compares the wall-clock time for transfer gain collection between our method and TAG on CelebA and ETTm1 (without any efficiency-enhancing strategies). Our full method incurs an overhead of approximately $1.87\times$ on CelebA (23.66 vs. 12.68 minutes) due to the additional joint-training forward passes; on ETTm1 our method is actually faster (175.01 vs. 188.17 minutes). Applying the lazy collection strategy at an interval of 50 steps reduces collection time by $\approx 16\times$ and $\approx 25\times$ respectively, bringing our wall-clock cost well below that of TAG on both datasets.

Table 3: Wall-clock time (minutes) for transfer gain collection. Lazy (50) denotes lazy collection with a 50-step interval; speedups are from Table 2.

| Method | CelebA (min) | ETTm1 (min) |
|---|---|---|
| TAG | 12.68 | 188.17 |
| Ours (full) | 23.66 | 175.01 |
| Ours (Lazy, step=50) | $\approx 1.48$ | $\approx 7.02$ |

**Lazy Collection Strategy.** We analyze how the frequency of transfer gain collection affects performance and computational efficiency across two datasets: CelebA and ETTm1. Table 2 presents relative improvements and computational speedups achieved with collection intervals ranging from 1 to 200 steps. For the CelebA dataset, relative improvement actually increases as collection frequency decreases (up to an interval of 50 steps). This counter-intuitive result suggests that less frequent collection may reduce estimation noise, leading to more robust task groupings. Beyond the 50-step threshold, performance begins to decline as transfer gain estimates become increasingly inaccurate due to insufficient sampling. The ETTm1 dataset exhibits a similar pattern, with optimal performance occurring at moderate collection intervals. These results demonstrate that our method can achieve substantial speedups ($10$–$15\times$) by employing a lazy collection strategy with intervals of 10–50 steps, while simultaneously maintaining or even enhancing performance. This finding has significant practical implications for deploying our method in compute-constrained environments.

We also investigate the impact of the number of epochs required for our proposed method on the final performance metrics. Figure 4 delineates the performance trends across different epochs for the CelebA and ETTm1 datasets. For CelebA, the total error exhibits a marked decrease as the number of epochs increases, stabilizing after approximately 20 epochs. This stabilization suggests that the model quickly benefits from

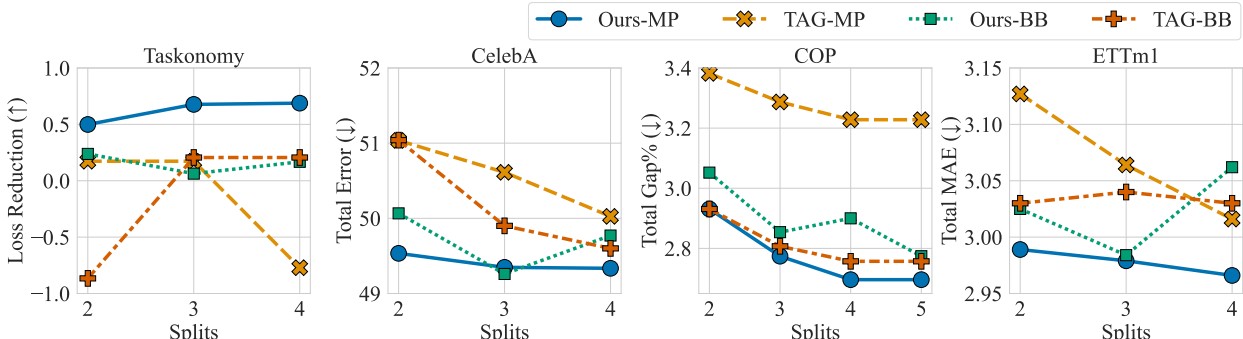

Figure 5: Performance comparison of Ours-MP, TAG-MP, Ours-BB and TAG-BB across multiple grouping splits on Taskonomy, CelebA, COP, and ETIm1 benchmarks.

initial training iterations but reaches a plateau, indicating little to no gain from additional training beyond this point. The shaded area. Conversely, the ETTm1 dataset shows a more gradual decline in total Mean Absolute Error (MAE) as the number of epochs grows. The initial drop in MAE is quite steep, suggesting significant learning gains from early training. Subsequently, the MAE curve flattens out after about 6 epochs, which implies that further training yields diminishing improvements in model performance. These findings demonstrate that the accumulation of transfer gains does not occur uniformly throughout all training periods; instead, it varies, with certain training phases yielding more substantial enhancements than others.

Combining the analysis, we can deduce several empirical guidelines for optimizing the collection of transfer gains with regard to efficiency: **(1)** The initial training period is crucial for uncovering task relationships, indicating the importance of concentrating resources on the early stages of training; **(2)** A range of 5-50 steps is considered optimal for gathering transfer gains, as it is probable that the gains from consecutive steps will be similar.

## 5.5 Ablation Study on Transfer Gain and Task Grouping Solver

In this section, we conduct an ablation study to highlight the significance of the novel transfer gain and the mathematical programming framework, compared to TAG (Fifty et al., 2021). To demonstrate the importance of our transfer gain metric, we apply the mathematical framework from Formulation 8 for task grouping using $\mathcal{S}_{i \to j}^t$ as per Formulation 3, and TAG's affinity $\mathcal{Z}_{i \to j}^t$. This approach is termed "Ours-MP" for our method and "TAG-MP" for TAG's method. We also evaluate our framework's performance against TAG's branch and bound techniques, introducing "Ours-BB" for the Branch & Bound method guided by the transfer gain $\mathcal{S}_{i \to j}^t$.

**Ablation on Transfer Gain.** The first aspect of the comparison focuses on the transfer gain between $\mathcal{S}_{i \to j}^t$ and $\mathcal{Z}_{i \to j}^t$, as implemented in Ours-MP and TAG-MP, respectively. Results in Figure 5 reveal that Ours-MP consistently surpasses TAG-MP across all benchmarks. This is evidenced by its greater loss reduction in the Taskonomy dataset, lower total

Table 4: Comparative Analysis of Time Efficiency Across Different Task Splits and Dataset. "s", "m" and "h" stand for seconds, minutes and hours, respectively. "×" indicates that the method fails to solve the problem within an 8-hour time limit.

| Splits | Taskonomy (5) | | CelebA (9) | | COP (6) | | ETTm1 (7) | |
|---|---|---|---|---|---|---|---|---|
| | BB | MP | BB | MP | BB | MP | BB | MP |
| 2 | 0.002s | 0.110s | 0.312s | 0.907s | 0.007s | 0.085s | 0.024s | 0.122s |
| 3 | 0.046s | 0.563s | 1.93m | 2.255s | 0.279s | 0.713s | 2.005s | 1.078s |
| 4 | 0.431s | 1.253s | × | 4.009s | 5.955s | 0.899s | 1.63m | 1.677s |
| 5 | 2.454s | 1.733s | × | 7.261s | 1.48m | 1.621s | 1.02h | 2.218s |
| 6 | - | - | × | 21.44s | 16.96m | 5.013s | × | 4.158s |

error in the CelebA dataset, minimized Total Gap percentage in COP, and reduced Total MAE in ETIm1 with an increasing number of grouping splits. TAG-MP's declining performance, particularly in the CelebA dataset, suggests that the transfer gain proposed in our methodology more accurately captures the task relationships than the one proposed in TAG, under a consistent task grouping solver.

**Ablation on Task Grouping Solver.** In Figure 5, Ours-MP demonstrates better performance than Ours-BB, providing results across all grouping splits.

We further demonstrate the detailed time-cost for Ours-MP and Ours-BB in Table 4, labeled as MP and BB, respectively. The results indicate that MP exhibits better time efficiency and scalability across different task splits and datasets compared to BB. Specifically, MP generally scales more effectively with an increasing number of splits than the baseline method BB. In particular, for CelebA, our method demonstrates exceptional coverage, capable of handling all scenarios up to six splits efficiently within 30 seconds. When examining the COP dataset, it is evident that the time cost for the BB method increases substantially, potentially exponentially, with larger splits, or fails to deliver results within an eight-hour limit. This trend is also observable across datasets; within the same split categories of two and three, the time cost for BB grows drastically with the increase in task numbers. This highlights our method's scalability and robustness in managing increased computational demands across varying scenarios.

## 6 Conclusions

In conclusion, our work presents a novel approach to task grouping in Multi-Task Learning (MTL), marking a significant advancement over existing methods. Our approach features two innovations: a robust transfer gains construction that operates without restrictive assumptions, and a flexible mathematical programming formulation tailored for task grouping challenges. Empirical results demonstrate our method's superiority, showing improved performance, flexibility, and efficiency in real-world settings, thus enhancing MTL models' applicability in diverse and resource-limited environments. We discuss the limitations and potential future research directions in Section F.

## Acknowledgments

This work was supported by the National Key R&D Program of China under grant 2022YFA1003900

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

## A  Proof of Proposition 1

*Proof.* Let $\tilde{L}_j(\phi) = L_j(\phi, \theta_j^{t+1})$ denote the loss function of task $j$ with $\theta_j$ fixed at step $t+1$. The group transfer gain from $A$ to $j$ is

$$\mathcal{S}_{A \to j}^t = 1 - \frac{\tilde{L}_j(\phi_{A \cup \{j\}}^{t+1})}{\tilde{L}_j(\phi_{\{j\}}^{t+1})} = \frac{\tilde{L}_j(\phi_{\{j\}}^{t+1}) - \tilde{L}_j(\phi_{A \cup \{j\}}^{t+1})}{\tilde{L}_j(\phi_{\{j\}}^{t+1})}.$$

Similarly, the task transfer gain from $i$ to $j$ is

$$\mathcal{S}_{i \to j}^t = 1 - \frac{\tilde{L}_j(\phi_{\{i,j\}}^{t+1})}{\tilde{L}_j(\phi_{\{j\}}^{t+1})} = \frac{\tilde{L}_j(\phi_{\{j\}}^{t+1}) - \tilde{L}_j(\phi_{\{i,j\}}^{t+1})}{\tilde{L}_j(\phi_{\{j\}}^{t+1})}.$$

We consider a standard one-step gradient update for the shared parameters $\phi$. Let $G_k^t = \nabla_\phi L_k(\phi^t, \theta_k^t)$ be the gradient of task $k$'s loss with respect to $\phi$ at step $t$. We assume the updates are given by:

$$\phi_{\{j\}}^{t+1} = \phi^t - \eta_t G_j^t$$
$$\phi_{\{i,j\}}^{t+1} = \phi^t - \eta_t (G_i^t + G_j^t)$$
$$\phi_{A \cup \{j\}}^{t+1} = \phi^t - \eta_t \left( \sum_{k \in A} G_k^t + G_j^t \right)$$

This assumes gradients are summed for shared parameters, other aggregation schemes are possible but this is common. Task-specific parameters $\theta_k$ are assumed to be updated based only on $L_k$, ensuring $\theta_j^{t+1}$ is the same across different scenarios for evaluating $L_j$.

The differences in shared parameters after one step, relative to $\phi_{\{j\}}^{t+1}$, are:

$$\phi_{\{i,j\}}^{t+1} - \phi_{\{j\}}^{t+1} = -\eta_t G_i^t$$
$$\phi_{A \cup \{j\}}^{t+1} - \phi_{\{j\}}^{t+1} = -\eta_t \sum_{k \in A} G_k^t$$

Assuming $\tilde{L}_j$ is differentiable, we can apply the Mean Value Theorem:

$$\tilde{L}_j(\phi_{\{j\}}^{t+1}) - \tilde{L}_j(\phi_{X \cup \{j\}}^{t+1}) = \nabla \tilde{L}_j(\xi_X)^T (\phi_{\{j\}}^{t+1} - \phi_{X \cup \{j\}}^{t+1}),$$

where $\xi_X$ is some point on the line segment between $\phi_{\{j\}}^{t+1}$ and $\phi_{X \cup \{j\}}^{t+1}$. Substituting the parameter differences:

$$\tilde{L}_j(\phi_{\{j\}}^{t+1}) - \tilde{L}_j(\phi_{\{i,j\}}^{t+1}) = \nabla \tilde{L}_j(\xi_i)^T (-\eta_t G_i^t) = -\eta_t \nabla \tilde{L}_j(\xi_i)^T G_i^t$$

$$\tilde{L}_j(\phi_{\{j\}}^{t+1}) - \tilde{L}_j(\phi_{A \cup \{j\}}^{t+1}) = \nabla \tilde{L}_j(\xi_A)^T (-\eta_t \sum_{k \in A} G_k^t) = -\eta_t \nabla \tilde{L}_j(\xi_A)^T \left( \sum_{k \in A} G_k^t \right)$$

where $\xi_i$ is between $\phi_{\{j\}}^{t+1}$ and $\phi_{\{i,j\}}^{t+1}$, and $\xi_A$ is between $\phi_{\{j\}}^{t+1}$ and $\phi_{A \cup \{j\}}^{t+1}$. Now, let's express the transfer gains using these results:

$$\mathcal{S}_{i \to j}^t = \frac{-\eta_t \nabla \tilde{L}_j(\xi_i)^T G_i^t}{\tilde{L}_j(\phi_{\{j\}}^{t+1})} \quad \text{and} \quad \mathcal{S}_{A \to j}^t = \frac{-\eta_t \nabla \tilde{L}_j(\xi_A)^T \left( \sum_{k \in A} G_k^t \right)}{\tilde{L}_j(\phi_{\{j\}}^{t+1})}.$$

Consider the difference:

$$\mathcal{S}_{A \to j}^t - \frac{1}{|A|} \sum_{i \in A} \mathcal{S}_{i \to j}^t = \frac{-\eta_t \nabla \tilde{L}_j(\xi_A)^T \left(\sum_{k \in A} G_k^t\right)}{\tilde{L}_j(\phi_{\{j\}}^{t+1})} - \frac{1}{|A|} \sum_{i \in A} \frac{-\eta_t \nabla \tilde{L}_j(\xi_i)^T G_i^t}{\tilde{L}_j(\phi_{\{j\}}^{t+1})}$$

$$= \frac{-\eta_t}{\tilde{L}_j(\phi_{\{j\}}^{t+1})} \left[ \nabla \tilde{L}_j(\xi_A)^T \left(\sum_{i \in A} G_i^t\right) - \frac{1}{|A|} \sum_{i \in A} \nabla \tilde{L}_j(\xi_i)^T G_i^t \right]$$

$$= \frac{-\eta_t}{\tilde{L}_j(\phi_{\{j\}}^{t+1})} \sum_{i \in A} \left[ \nabla \tilde{L}_j(\xi_A)^T G_i^t - \frac{1}{|A|} \nabla \tilde{L}_j(\xi_i)^T G_i^t \right]$$

$$= \frac{-\eta_t}{\tilde{L}_j(\phi_{\{j\}}^{t+1})} \sum_{i \in A} \left( \nabla \tilde{L}_j(\xi_A) - \frac{1}{|A|} \nabla \tilde{L}_j(\xi_i) \right)^T G_i^t.$$

Taking the absolute value and using the assumption $0 < C \le L_j$ for the denominator $\tilde{L}_j(\phi_{\{j\}}^{t+1})$:

$$\left| \mathcal{S}_{A \to j}^t - \frac{1}{|A|} \sum_{i \in A} \mathcal{S}_{i \to j}^t \right| \le \frac{\eta_t}{C} \left| \sum_{i \in A} \left( \nabla \tilde{L}_j(\xi_A) - \frac{1}{|A|} \nabla \tilde{L}_j(\xi_i) \right)^T G_i^t \right|.$$

By the triangle inequality and Cauchy-Schwarz inequality:

$$\le \frac{\eta_t}{C} \sum_{i \in A} \left| \left( \nabla \tilde{L}_j(\xi_A) - \frac{1}{|A|} \nabla \tilde{L}_j(\xi_i) \right)^T G_i^t \right| \le \frac{\eta_t}{C} \sum_{i \in A} ||\nabla \tilde{L}_j(\xi_A) - \frac{1}{|A|} \nabla \tilde{L}_j(\xi_i)|| \cdot ||G_i^t||.$$

Since $L_k$ is $l$-Lipschitz for all $k$, its gradient with respect to $\phi$ is bounded by $l$, i.e., $||\nabla_\phi L_k(\phi, \theta_k)|| \le l$ for all $\phi, \theta_k$. Thus, $||G_i^t|| = ||\nabla_\phi L_i(\phi^t, \theta_i^t)|| \le l$. Also, $||\nabla \tilde{L}_j(\phi)|| = ||\nabla_\phi L_j(\phi, \theta_j^{t+1})|| \le l$ for any $\phi$. Using the triangle inequality for the gradient difference:

$$||\nabla \tilde{L}_j(\xi_A) - \frac{1}{|A|} \nabla \tilde{L}_j(\xi_i)|| \le ||\nabla \tilde{L}_j(\xi_A)|| + \frac{1}{|A|} ||\nabla \tilde{L}_j(\xi_i)|| \le l + \frac{1}{|A|} l = l \left( 1 + \frac{1}{|A|} \right).$$

Substituting these bounds:

$$\le \frac{\eta_t}{C} \sum_{i \in A} l \left( 1 + \frac{1}{|A|} \right) \cdot l = \frac{\eta_t l^2}{C} \sum_{i \in A} \left( 1 + \frac{1}{|A|} \right) = \frac{\eta_t l^2}{C} \left( |A| \cdot 1 + |A| \cdot \frac{1}{|A|} \right) = \frac{\eta_t l^2}{C} (|A| + 1).$$

This completes the proof. $\square$

## B    Experimental Details

This section provides a comprehensive overview of the experimental settings and results, including the datasets descriptions, model architectures used, hyperparameters, benchmark methods, and modifications implemented to ensure a more equitable comparison.

### B.1    Datasets Descriptions

**Taskonomy.** Taskonomy is a comprehensive dataset designed to facilitate systematic studies of the relationships among visual tasks. It consists of over 4.5 million images gathered from more than 4,000 indoor scenes. These images are annotated for 26 different tasks including depth prediction, surface normal estimation, and semantic segmentation. Taskonomy aims to support research in multitask learning and transfer learning by providing a wide range of task annotations. It supports five primary vision tasks: Semantic Segmentation (s), Depth Estimation (d), Surface Normal (n), Keypoint Detection (k), and Canny Edge Detection (e). In accordance with the standard criterion utilized in previous studies, these five tasks take precedence in the conduction of experiments.

**CelebA.** CelebA is a large-scale face attributes dataset, which is widely used for multitask learning involving facial attribute recognition. It contains over 200,000 images of 10,000 celebrities, each annotated with 40 attribute labels such as "Smiling", "Young", "Male", and "Wearing Hat". Following the protocol outlined in TAG (Fifty et al., 2021), we select a subset of 9 attributes: 5 o'Clock Shadow, Black Hair, Blond Hair, Brown Hair, Goatee, Mustache, No Beard, Rosy Cheeks, and Wearing Hat, denoted as $\{a_1, a_2, a_3, a_4, a_5, a_6, a_7, a_8, a_9\}$, from the original 40 attributes for experimental purposes.

**Combinatorial Optimization Benchmarks.** We explore three types of COPs: the Travelling Salesman Problem (TSP), the Capacitated Vehicle Routing Problem (CVRP) and the Orienteering Problem (OP). Two problem scales are considered for each COP: 20 and 50 for TSP, CVRP, and OP. We employ the notation "COP-scale", such as TSP-20, to denote a particular task, resulting in a total of 6 tasks.

**ETTm1.** The ETTm1 dataset comprises detailed recordings from an electricity transformer situated in a specific region of China, spanning data from July 2016 to July 2018. This dataset encompasses six power load series, along with an oil temperature series, recorded at 15-minute intervals. These series are classified as High Useful load (HF), High Useless load (HL), Middle Useful load (MF), Middle Useless load (ML), Low Useful load (LF), and Low Useless load (LL). The dataset is utilized for forecasting purposes, leveraging all series as inputs and focusing on predictions for individual series as separate tasks. Following the settings in MTG (Song et al., 2022), it is split into training, validation, and test sets following a 6:2:2 chronological order ratio.

## B.2 Experimental Settings

**Backbones and hyper-parameters.** In the computer vision tasks, Taskonomy and CelebA, we employ a simplified ResNet18 encoder coupled with MLP decoders for each task. Model structure hyperparameters and training attributes such as hidden dimensions, encoder layers, initial learning rate, and scheduling method mirror those in Fifty et al. (2021). For combinatorial optimization benchmarks, we use POMO (Kwon et al., 2020) as the backbone, with all hyperparameters held constant except for the training episodes, which are set to 10,000. In the domain of time series analysis, the Autoformer architecture is employed as the neural network structure. Time series forecasting encompasses two widely recognized approaches: multivariate and univariate. Given that the ETTm1 dataset comprises seven time series, it is applicable under both frameworks. To facilitate a task grouping experiment, we configure it as several univariate prediction tasks, adapting the Autoformer model to maintain a majority of its components common across tasks while assigning a unique decoder to each task for making predictions. Regarding the detailed hyperparameter settings for the model's structure and training, we adhere to the configurations specified by Song et al. (2022).

**Baselines.** Our experimental evaluation involves a comprehensive comparison against a range of established methods: (1) Single Task Learning (STL); (2) MTL methods: We consider a variety of MTL methods that employ different strategies for joint task learning, including: Naive-MTL, Bandit-MTL (Mao et al., 2021), PCGrad (Yu et al., 2020), Nash-MTL (Navon et al., 2022), Uncertainty-Weighting (UW) (Kendall et al., 2018) and LinearScale; (3) Task grouping methods: Random policy by which tasks are grouped randomly and results are taken the average for 10 repeats; Optimal policy which is obtained by enumeration; TAG (Fifty et al., 2021), a known state-of-the-art (SOTA) task grouping method to group tasks based on their affinity. We also evaluate the Linear Surrogate (Li et al., 2023), which fits a linear surrogate to MTL validation losses from randomly sampled source subsets to estimate per-task relevance and selects sources via thresholding to mitigate negative transfer. Additionally, we include an extra high order approximation (HOA) (Standley et al., 2020) and cosine similarity (CS) in Taskonomy and CelebA and MTG (Song et al., 2022) in Taskonomy and ETTm1. We further include two recent task-grouping baselines, Selective Task Group Updates (STG) (Jeong & Yoon, 2025) and Grad-TAG (Li et al., 2024), which respectively perform selective task-set updates within a shared network and estimate task affinity from gradients for scalable clustering.

**Equipments.** The experiments were conducted on a server equipped with 8 NVIDIA A100 Tensor Core GPU and 128 Intel(R) Xeon(R) Platinum 8358P CPUs. The primary software versions used are CUDA 11.8, TensorFlow 2.14.1, and PyTorch 2.1.2.

Table 5: Grouing and comparison results on the Taskonomy dataset.

| | Method | s | d | n | k | e | Loss Reduction (↑) |
|---|---|---|---|---|---|---|---|
| | Naive-MTL | 0.138 | 0.088 | −0.028 | −0.077 | 0.052 | 0.173 |
| | Surrogate | −0.033 | 0.088 | −0.028 | −0.029 | 0.052 | 0.050 |
| **2 Groups** | Random | −0.033 | 0.049 | 0.000 | −0.043 | 0.043 | 0.016 |
| | Optimal | - | - | −0.148 | 0.443 | - | 0.694 |
| | | 0.138 | 0.088 | −0.028 | −0.077 | 0.052 | |
| | CS | −0.008 | −0.690 | −0.167 | - | - | −0.866 |
| | | - | - | - | −0.013 | 0.012 | |
| | HOA | 0.072 | 0.014 | −0.002 | - | 0.080 | 0.608 |
| | | - | - | −0.148 | 0.443 | - | |
| | TAG | −0.008 | −0.690 | −0.167 | - | - | −0.866 |
| | | - | - | - | −0.013 | 0.012 | |
| | MTG | - | 0.031 | −0.068 | 0.249 | 0.032 | 0.499 |
| | | 0.138 | 0.088 | −0.028 | −0.077 | 0.052 | |
| | Ours | 0.138 | 0.088 | −0.028 | −0.077 | 0.052 | 0.499 |
| | | - | 0.031 | −0.068 | 0.249 | 0.032 | |
| **3 Groups** | Random | 0.149 | 0.020 | −0.062 | 0.062 | 0.020 | 0.190 |
| | Optimal | 0.199 | - | −0.101 | - | - | 0.888 |
| | | - | - | −0.148 | 0.443 | - | |
| | | - | 0.117 | 0.018 | - | 0.110 | |
| | CS | 0.052 | 0.020 | - | - | - | 0.206 |
| | | - | 0.146 | 0.008 | - | - | |
| | | - | - | - | −0.013 | 0.012 | |
| | HOA | −0.008 | −0.690 | −0.167 | - | - | −0.391 |
| | | - | - | −0.148 | 0.443 | - | |
| | | - | - | - | −0.013 | 0.012 | |
| | TAG | 0.052 | 0.020 | - | - | - | 0.206 |
| | | - | 0.146 | 0.008 | - | - | |
| | | - | - | - | −0.013 | 0.012 | |
| | MTG | - | 0.146 | 0.008 | - | - | 0.598 |
| | | - | −0.013 | - | 0.247 | 0.058 | |
| | | 0.138 | 0.088 | −0.028 | −0.077 | 0.052 | |
| | Ours | - | - | −0.101 | 0.412 | 0.067 | 0.677 |
| | | - | 0.031 | −0.068 | 0.249 | 0.032 | |
| | | 0.138 | 0.088 | −0.028 | −0.077 | 0.052 | |
| **4 Groups** | Random | 0.063 | 0.031 | −0.068 | 0.249 | 0.032 | 0.307 |
| | Optimal | - | - | 0.000 | - | - | 0.833 |
| | | 0.199 | - | −0.101 | - | - | |
| | | - | 0.146 | 0.008 | - | - | |
| | | - | - | −0.101 | 0.412 | 0.067 | |
| | CS | 0.052 | 0.020 | - | - | - | 0.206 |
| | | - | 0.146 | 0.008 | - | - | |
| | | −0.008 | −0.690 | −0.167 | - | - | |
| | | - | - | - | −0.013 | 0.012 | |
| | HOA | 0.199 | - | −0.101 | - | - | 0.809 |
| | | - | 0.146 | 0.008 | - | - | |
| | | - | - | −0.148 | 0.443 | - | |
| | | - | - | - | −0.013 | 0.012 | |
| | TAG | 0.052 | 0.020 | - | - | - | 0.206 |
| | | - | 0.146 | 0.008 | - | - | |
| | | −0.008 | −0.690 | −0.167 | - | - | |
| | | - | - | - | −0.013 | 0.012 | |
| | MTG | 0.000 | - | - | - | - | 0.598 |
| | | - | 0.146 | 0.008 | - | - | |
| | | - | −0.013 | - | 0.247 | 0.058 | |
| | | 0.138 | 0.088 | −0.028 | −0.077 | 0.052 | |
| | Ours | - | - | −0.101 | 0.412 | 0.067 | 0.688 |
| | | 0.138 | 0.088 | −0.028 | −0.077 | 0.052 | |
| | | - | 0.031 | −0.068 | 0.249 | 0.032 | |
| | | 0.149 | - | −0.062 | 0.062 | 0.020 | |

### B.3   Further Results on Taskonomy Dataset

We compare the proposed methods with Naive-MTL, Random baseline, Optimal baseline, HOA, CS and TAG in Table 5. The key metric for evaluation is the loss reduction borrowing the results from MTG (Song et al., 2022), where a higher value indicates better performance.

From the results in Table 5, the Naive-MTL method shows a moderate performance with a loss reduction of 0.173. The Linear Surrogate baseline (Li et al., 2023) yields only a modest gain of 0.050—noticeably below Naive-MTL—suggesting limited effectiveness on Taskonomy under our setup.

When splitting 2 groups, the Random method, CS and TAG perform poorly, resulting in a significant negative loss reduction of $-0.866$. The HOA method achieves the best results among non-oracle baselines (0.608), while the proposed method (Ours) attains the second-best performance after the Optimal baseline (0.694) with a notable loss reduction of 0.499.

For the 3 Groups category, the proposed method (Ours) performs strongly, again ranking behind only the Optimal baseline (0.888), with a loss reduction of 0.677. In contrast, HOA becomes unstable and performs the worst among all baselines with a negative loss reduction.

As the number of groups increases (e.g., 4 groups), the proposed method maintains competitive performance (0.688), second only to the Optimal baseline (0.833), and surpasses Random (0.307), CS/TAG (0.206), and MTG (0.598). Overall, our method shows progressive improvements and robustness across different group divisions, whereas HOA—despite excelling in the 2-group case—exhibits instability as the group count changes.

### B.4   Further Results on CelebA Dataset

We compare our approach against four benchmark MTL methodologies, namely, Naive-MTL, GradNorm, PCGrad, and Uncertainty Weights. Notably, we directly implement the precise groupings presented by Fifty et al. (2021) as the grouping outcome of TAG, HOA, and CS methods, though the total error is based on our testing outcomes. For completeness, we also include the Linear Surrogate (Surrogate) baseline, which attains a total error of 51.000—better than GradNorm (51.312) and UW (52.717), comparable to PCGrad (50.835), but still behind Naive-MTL (50.233) and all of our grouping-based variants; this suggests that a simple linear proxy for source selection is less effective on CelebA than explicit task grouping. In addition, the recent STG baseline yields a markedly higher total error of 55.030 (worse than STL), indicating severe negative transfer on CelebA, while Grad-TAG variants remain close to Naive-MTL (total error in the range 50.39–50.95) but still lag behind our grouping-based methods and the OPT oracle, and their SDP-based rounding cannot realize some required split configurations (e.g., valid 4-way partitions on CelebA), further limiting their applicability in our setting.

As Table 6 exhibits, our method typically surpasses other grouping methodologies and MTL methods, barring OPT groupings derived from TAG. These may not necessarily represent the optimal groupings in our test results, as our approach outperforms OPT in the 2-split scenario, where our approach achieves a total error of 49.534, with OPT at 49.583. For the 3-split scenario, our approach achieves better results of 49.348 compared to the 2-split case, reaching almost the same performance as the OPT grouping. Furthermore, it continues to exhibit improvement with the 4-split case, achieving a total error of 49.335. As the number of groups increases, TAG and our proposed method demonstrate continuous improvement, while HOA performance declines, and CS exhibits instability.

### B.5   Further Results on Combinatorial Optimization Tasks

In the presented comparative analysis in Table 7, Single-Task Learning (STL) demonstrates a strong baseline with a total gap of 2.876%, outperforming all Multi-Task Learning (MTL) methods in terms of the Total Gap metric. Specifically, STL exhibits a lower total gap compared to the best-performing MTL methods, UW and the Linear Surrogate (Surrogate), both at 3.007%. Notably, the Surrogate model is competitive within MTL—matching UW in total gap while slightly improving on CVRP20/50 and OP20—yet it still trails STL overall.

Table 6: Grouing and comparison results on the CelebA dataset.

1

| | Method | a1 | a2 | a3 | a4 | a5 | a6 | a7 | a8 | a9 | Tot. Error (↓) |
|---|---|---|---|---|---|---|---|---|---|---|---|
| | STL | 6.47 ± 0.044 | 11.27 ± 0.037 | 4.19 ± 0.006 | 12.29 ± 0.020 | 2.72 ± 0.038 | 3.12 ± 0.015 | 4.98 ± 0.031 | 4.85 ± 0.019 | 0.73 ± 0.007 | 50.621 |
| MTL | Naive-MTL | 6.55 ± 0.016 | 11.09 ± 0.023 | 4.19 ± 0.014 | 12.56 ± 0.101 | 2.58 ± 0.015 | 3.02 ± 0.027 | 4.80 ± 0.017 | 4.74 ± 0.034 | 0.70 ± 0.004 | 50.233 |
| | GradNorm | 7.18 ± 0.071 | 11.35 ± 0.028 | 4.21 ± 0.034 | 12.18 ± 0.127 | 2.52 ± 0.015 | 2.85 ± 0.032 | 5.01 ± 0.084 | 5.29 ± 0.103 | 0.72 ± 0.019 | 51.312 |
| | PCGrad | 6.58 ± 0.101 | 11.12 ± 0.019 | 4.27 ± 0.048 | 12.67 ± 0.722 | 2.61 ± 0.034 | 2.92 ± 0.010 | 4.96 ± 0.204 | 5.02 ± 0.021 | 0.69 ± 0.030 | 50.835 |
| | UW | 6.72 ± 0.037 | 11.32 ± 0.019 | 4.30 ± 0.150 | 13.61 ± 0.541 | 2.74 ± 0.049 | 2.93 ± 0.051 | 5.43 ± 0.175 | 4.86 ± 0.040 | 0.79 ± 0.025 | 52.717 |
| | Surrogate | 6.55 ± 0.016 | 11.30 ± 0.082 | 4.19 ± 0.014 | 12.56 ± 0.101 | 2.68 ± 0.004 | 3.02 ± 0.027 | 5.06 ± 0.012 | 4.95 ± 0.017 | 0.70 ± 0.004 | 51.000 |
| | STG | 7.00 ± 0.008 | 11.50 ± 0.013 | 4.62 ± 0.017 | 12.53 ± 0.018 | 3.20 ± 0.016 | 3.52 ± 0.017 | 5.32 ± 0.028 | 5.33 ± 0.014 | 1.17 ± 0.008 | 54.200 |
| 2 Groups | Random | 6.41 ± 0.031 | 11.10 ± 0.019 | 4.14 ± 0.009 | 12.51 ± 0.033 | 2.66 ± 0.007 | 2.92 ± 0.017 | 4.71 ± 0.027 | 4.75 ± 0.025 | 0.78 ± 0.011 | 49.987 |
| | OPT | - | 11.16 ± 0.002 | 4.04 ± 0.028 | 12.24 ± 0.367 | - | - | - | - | 0.74 ± 0.025 | 49.583 |
| | | 6.60 ± 0.017 | 11.22 ± 0.042 | 4.36 ± 0.018 | 12.05 ± 0.133 | 2.60 ± 0.010 | 2.88 ± 0.011 | 4.82 ± 0.037 | 4.71 ± 0.037 | - | |
| | TAG | 6.50 ± 0.158 | - | - | - | - | - | 4.88 ± 0.118 | - | - | 51.036 |
| | | - | 11.16 ± 0.053 | 4.24 ± 0.038 | 13.15 ± 0.112 | 2.66 ± 0.004 | 2.99 ± 0.006 | - | 4.74 ± 0.010 | 0.71 ± 0.005 | |
| | CS | - | - | - | - | 2.59 ± 0.026 | 3.06 ± 0.017 | - | - | - | 50.278 |
| | | 6.58 ± 0.028 | 11.15 ± 0.027 | 4.26 ± 0.097 | 12.35 ± 0.229 | 2.54 ± 0.029 | - | 4.86 ± 0.059 | 4.77 ± 0.083 | 0.71 ± 0.017 | |
| | HOA | - | - | - | - | 2.57 ± 0.019 | - | 4.69 ± 0.024 | - | - | 49.600 |
| | | 6.55 ± 0.018 | 11.27 ± 0.026 | 4.14 ± 0.009 | 11.87 ± 0.045 | - | 3.03 ± 0.016 | - | 4.81 ± 0.025 | 0.67 ± 0.006 | |
| | Grad-TAG | 6.45 ± 0.007 | - | 4.19 ± 0.061 | 12.25 ± 0.037 | 2.80 ± 0.024 | 2.97 ± 0.006 | - | 4.93 ± 0.039 | 0.77 ± 0.005 | 50.393 |
| | | - | 11.24 ± 0.099 | - | - | - | - | 4.80 ± 0.083 | - | - | |
| | Ours | 6.55 ± 0.016 | 11.09 ± 0.023 | 4.19 ± 0.014 | 12.56 ± 0.101 | 2.58 ± 0.015 | 3.02 ± 0.027 | 4.80 ± 0.017 | 4.74 ± 0.034 | 0.70 ± 0.004 | 49.534 |
| | | 6.60 ± 0.017 | 11.22 ± 0.042 | 4.36 ± 0.018 | 12.05 ± 0.133 | 2.60 ± 0.010 | 2.88 ± 0.011 | 4.82 ± 0.037 | 4.71 ± 0.037 | - | |
| 3 Groups | Random | 6.65 ± 0.030 | 11.16 ± 0.018 | 4.12 ± 0.019 | 12.16 ± 0.147 | 2.61 ± 0.065 | 2.90 ± 0.019 | 4.78 ± 0.018 | 4.71 ± 0.010 | 0.79 ± 0.012 | 49.886 |
| | OPT | 6.30 ± 0.036 | - | - | - | - | - | 4.67 ± 0.013 | - | 0.71 ± 0.009 | 49.323 |
| | | - | 11.16 ± 0.002 | 4.04 ± 0.028 | 12.24 ± 0.367 | - | - | - | - | 0.74 ± 0.025 | |
| | | 6.27 ± 0.027 | - | - | - | 2.59 ± 0.006 | 2.92 ± 0.011 | 4.73 ± 0.016 | 4.72 ± 0.019 | 0.76 ± 0.002 | |
| | TAG | 6.50 ± 0.158 | - | - | - | - | - | 4.88 ± 0.118 | - | - | 49.897 |
| | | - | 11.14 ± 0.047 | - | 12.31 ± 0.030 | - | - | - | - | - | |
| | | - | 11.18 ± 0.160 | 4.11 ± 0.007 | - | 2.55 ± 0.029 | 2.92 ± 0.063 | 4.95 ± 0.117 | 4.73 ± 0.013 | 0.76 ± 0.014 | |
| | CS | - | - | - | - | 2.59 ± 0.026 | 3.06 ± 0.017 | - | - | - | 50.954 |
| | | 6.50 ± 0.158 | - | - | - | - | - | 4.88 ± 0.118 | - | - | |
| | | - | 11.18 ± 0.008 | 4.10 ± 0.013 | 13.06 ± 0.105 | - | - | - | 4.83 ± 0.013 | 0.74 ± 0.004 | |
| | HOA | - | - | - | - | 2.57 ± 0.019 | - | 4.69 ± 0.024 | - | - | 49.716 |
| | | - | 11.14 ± 0.047 | - | 12.31 ± 0.030 | - | - | - | - | - | |
| | | 6.55 ± 0.024 | - | 4.12 ± 0.023 | - | 2.70 ± 0.006 | 2.84 ± 0.002 | 4.89 ± 0.021 | 4.76 ± 0.011 | 0.73 ± 0.012 | |
| | Grad-TAG | 6.60 ± 0.030 | - | - | 12.41 ± 0.141 | - | 3.02 ± 0.014 | - | - | 0.75 ± 0.016 | 50.521 |
| | | - | 11.15 ± 0.031 | - | - | - | - | 4.94 ± 0.019 | 4.78 ± 0.017 | - | |
| | | - | - | 4.17 ± 0.057 | - | 2.70 ± 0.009 | - | - | - | - | |
| | Ours | 6.60 ± 0.017 | 11.22 ± 0.042 | 4.36 ± 0.018 | 12.05 ± 0.133 | 2.60 ± 0.010 | 2.88 ± 0.011 | 4.82 ± 0.037 | 4.71 ± 0.037 | - | 49.348 |
| | | 6.55 ± 0.016 | 11.09 ± 0.023 | 4.19 ± 0.014 | 12.56 ± 0.101 | 2.58 ± 0.015 | 3.02 ± 0.027 | 4.80 ± 0.017 | 4.74 ± 0.034 | 0.70 ± 0.004 | |
| | | 6.62 ± 0.043 | 11.32 ± 0.131 | 4.09 ± 0.015 | 12.21 ± 0.051 | 2.60 ± 0.044 | 2.93 ± 0.008 | 4.72 ± 0.053 | - | - | |
| 4 Groups | Random | 6.40 ± 0.023 | 11.25 ± 0.029 | 4.11 ± 0.010 | 12.92 ± 0.102 | 2.55 ± 0.014 | 3.01 ± 0.002 | 4.74 ± 0.023 | 4.75 ± 0.006 | 0.76 ± 0.013 | 50.493 |
| | OPT | - | - | 4.08 ± 0.079 | - | - | 2.94 ± 0.021 | 4.82 ± 0.033 | - | 0.74 ± 0.006 | 49.169 |
| | | - | 11.29 ± 0.129 | 4.12 ± 0.056 | 12.59 ± 0.090 | - | - | - | - | - | |
| | | 6.30 ± 0.036 | - | - | - | - | - | 4.67 ± 0.013 | - | 0.71 ± 0.009 | |
| | | - | 11.11 ± 0.020 | - | 11.92 ± 0.012 | 2.68 ± 0.019 | - | - | 4.77 ± 0.006 | 0.78 ± 0.004 | |
| | TAG | 6.50 ± 0.158 | - | - | - | - | - | 4.88 ± 0.118 | - | - | 49.605 |
| | | - | 11.14 ± 0.047 | - | 12.31 ± 0.030 | - | - | - | - | - | |
| | | - | - | - | - | 2.62 ± 0.010 | 2.99 ± 0.004 | 4.76 ± 0.009 | - | - | |
| | | - | 10.97 ± 0.023 | 4.09 ± 0.033 | 12.40 ± 0.054 | - | - | 4.95 ± 0.017 | 4.63 ± 0.012 | 0.72 ± 0.002 | |
| | CS | - | - | - | - | 2.59 ± 0.026 | 3.06 ± 0.017 | - | - | - | 49.753 |
| | | 6.50 ± 0.158 | - | - | - | - | - | 4.88 ± 0.118 | - | - | |
| | | - | 11.14 ± 0.047 | - | 12.31 ± 0.030 | - | - | - | - | - | |
| | | - | 10.97 ± 0.023 | 4.09 ± 0.033 | 12.40 ± 0.054 | - | - | 4.95 ± 0.017 | 4.63 ± 0.012 | 0.72 ± 0.002 | |
| | HOA | - | - | - | - | 2.57 ± 0.019 | - | 4.69 ± 0.024 | - | - | 49.853 |
| | | - | 11.14 ± 0.047 | - | 12.31 ± 0.030 | - | - | - | - | - | |
| | | 6.65 ± 0.033 | - | - | - | 2.68 ± 0.031 | - | - | - | - | |
| | | - | 11.18 ± 0.160 | 4.11 ± 0.007 | - | 2.55 ± 0.029 | 2.92 ± 0.063 | 4.95 ± 0.117 | 4.73 ± 0.013 | 0.76 ± 0.014 | |
| | Ours | 6.62 ± 0.043 | 11.32 ± 0.131 | 4.09 ± 0.015 | 12.21 ± 0.051 | 2.60 ± 0.044 | 2.93 ± 0.008 | 4.72 ± 0.053 | - | - | 49.335 |
| | | 6.63 ± 0.026 | 11.36 ± 0.083 | 4.34 ± 0.034 | 12.23 ± 0.057 | 2.77 ± 0.019 | - | 4.94 ± 0.009 | 4.69 ± 0.027 | - | |
| | | 6.55 ± 0.016 | 11.09 ± 0.023 | 4.19 ± 0.014 | 12.56 ± 0.101 | 2.58 ± 0.015 | 3.02 ± 0.027 | 4.80 ± 0.017 | 4.74 ± 0.034 | 0.70 ± 0.004 | |
| | | 6.60 ± 0.017 | 11.22 ± 0.042 | 4.36 ± 0.018 | 12.05 ± 0.133 | 2.60 ± 0.010 | 2.88 ± 0.011 | 4.82 ± 0.037 | 4.71 ± 0.037 | - | |

Within the domain of Task Grouping methods, both TAG and our proposed method demonstrate the ability to surpass the STL baseline in certain aspects. Notably, our method consistently achieves the highest performance among non-optimal baselines across each grouping strategy, indicating its efficacy in handling

Table 7: Grouing and comparison results on the COP benchmark.

| | Method | TSP20 | TSP50 | CVRP20 | CVRP50 | OP20 | OP50 | Tot. Gap (↓) |
|---|---|---|---|---|---|---|---|---|
| | STL | 0.017% | 0.277% | 0.534% | 1.780% | −0.849% | 1.117% | 2.876% |
| MTL | Naive-MTL | 0.022% | 0.469% | 0.522% | 2.070% | −0.805% | 1.270% | 3.548% |
| | Bandit-MTL | 0.021% | 0.882% | 0.690% | 2.511% | −0.865% | 2.114% | 5.354% |
| | PCGrad | 0.028% | 0.708% | 0.605% | 2.411% | −0.689% | 1.756% | 4.819% |
| | UW | 0.042% | 0.362% | 0.412% | 1.703% | −0.665% | 1.153% | 3.007% |
| | LS | 0.020% | 0.476% | 0.512% | 2.084% | −0.792% | 1.197% | 3.498% |
| | Nash-MTL | 0.038% | 0.322% | 0.421% | 1.847% | −0.873% | 1.279% | 3.034% |
| | Surrogate | 0.022% | 0.388% | 0.405% | 1.692% | −0.771% | 1.270% | 3.007% |
| | STG | 0.020% | 0.728% | 0.674% | 2.355% | −0.457% | 1.582% | 4.901% |
| 3 Groups | Random | 0.031% | 0.535% | 0.416% | 1.741% | −0.764% | 1.279% | 3.237% |
| | OPT | - | - | 0.404% | 1.692% | - | - | 2.492% |
| | | 0.022% | 0.307% | 0.512% | - | - | - | |
| | | - | 0.638% | 0.631% | - | −0.861% | 0.929% | |
| | TAG | 0.021% | 0.324% | - | - | - | - | 2.807% |
| | | - | 0.505% | 0.405% | 1.724% | - | - | |
| | | - | - | - | - | −0.771% | 1.104% | |
| | Grad-TAG | 0.018% | - | - | - | −0.788% | - | 3.560% |
| | | - | 0.458% | 0.512% | - | - | - | |
| | | - | - | - | 2.252% | - | 1.108% | |
| | Ours | 0.021% | 0.324% | - | - | - | - | 2.774% |
| | | - | - | 0.404% | 1.692% | - | - | |
| | | - | - | - | - | −0.771% | 1.104% | |
| 4 Groups | Random | 0.024% | 0.473% | 0.681% | 2.125% | −0.852% | 1.072% | 3.522% |
| | OPT | - | 0.277% | - | - | - | - | 2.431% |
| | | - | - | 0.404% | 1.692% | - | - | |
| | | 0.027% | 0.415% | - | - | −0.897% | 1.120% | |
| | | - | 0.638% | 0.631% | - | −0.861% | 0.929% | |
| | TAG | 0.021% | 0.324% | - | - | - | - | 2.774% |
| | | - | 0.458% | 0.512% | - | - | - | |
| | | - | - | 0.404% | 1.692% | - | - | |
| | | - | - | - | - | −0.771% | 1.104% | |
| | Ours | 0.021% | 0.324% | - | - | - | - | 2.696% |
| | | - | - | 0.404% | 1.692% | - | - | |
| | | 0.034% | - | - | - | −0.840% | 1.095% | |
| | | - | - | - | - | −0.771% | 1.104% | |
| 5 Groups | Random | 0.038% | 0.444% | 0.472% | 1.992% | −0.761% | 1.233% | 3.419% |
| | OPT | 0.017% | - | - | - | - | - | 2.422% |
| | | - | 0.277% | - | - | - | - | |
| | | - | - | 0.404% | 1.692% | - | - | |
| | | 0.027% | 0.415% | - | - | −0.897% | 1.120% | |
| | | - | 0.638% | 0.631% | - | −0.861% | 0.929% | |
| | TAG | 0.021% | 0.324% | - | - | - | - | 2.757% |
| | | 0.022% | 0.307% | 0.512% | - | - | - | |
| | | - | 0.458% | 0.512% | - | - | - | |
| | | - | - | 0.404% | 1.692% | - | - | |
| | | - | - | - | - | −0.771% | 1.104% | |
| | Grad-TAG | 0.018% | - | - | - | −0.788% | - | 2.938% |
| | | - | 0.277% | - | - | - | - | |
| | | - | - | 0.534% | - | - | - | |
| | | - | - | - | 1.780% | - | - | |
| | | - | - | - | - | - | 1.117% | |
| | Ours | 0.021% | 0.324% | - | - | - | - | 2.696% |
| | | - | - | - | - | −0.771% | 1.104% | |
| | | - | - | 0.404% | 1.692% | - | - | |
| | | - | - | - | - | - | 1.117% | |
| | | 0.034% | - | - | - | −0.840% | 1.095% | |

multiple related tasks simultaneously. In contrast, the recent STG baseline attains a much larger total gap of 5.136%, reflecting severe negative transfer on COP, while Grad-TAG variants obtain total gaps in the range 2.94%–3.63%—roughly on par with Naive-MTL but still worse than STL and clearly behind our method and the OPT oracle.

As we consider the performance trends across different task groupings, it is observed that the efficacy of Optimal, TAG, and our method improves as the splits become larger. For instance, in the 3-split task grouping, our method achieves a total gap of 2.774%, compared to the optimal group's total gap of 2.492%. This trend suggests that the tasks within the COP benchmark exhibit high positive transfer potential.

In the context of 3-split task grouping, our method achieves logical groupings of tasks, pairing the same types of COPs together: (TSP20, TSP50), (CVRP20, CVRP50), and (OP20, OP50). This reflects an intuitive understanding that tasks of the same types benefit from being trained in concert. Intriguingly, the optimal groupings—(TSP20, TSP50, CVRP20), (CVRP20, CVRP50), and (TSP50, CVRP20, OP20, OP50)—do not align with these intuitive pairings, suggesting that there may be non-obvious correlations that, when leveraged, could lead to even greater improvements in task performance.

### B.6 Further Results on Time Series Tasks

Table 8 reveals that our approach outperforms other task grouping strategies across all divisions. Among the multitask learning (MTL) methods evaluated, PCGrad exhibits the best performance with an MAE of 2.997, outperforming other strategies such as Bandit-MTL (3.052) and Naive-MTL (3.038). The Linear Surrogate (Surrogate) achieves an MAE of 3.040—essentially on par with Naive-MTL and slightly better than STL (3.050)—indicating only modest benefit from its surrogate selection on ETTm1 under our setup. However, these MTL methods still yield higher MAE values compared to our approach.

Three task grouping strategies were examined: Random, OPT, and TAG. The Random strategy achieves an MAE of 3.035, comparable to Naive-MTL. OPT exhibits the best performance in the 2-split scenario, with an MAE of 2.926, while TAG performs comparably to Random, with an MAE of 3.032. Beyond these baselines, the STG method attains an MAE of 3.065, slightly worse than STL, whereas Grad-TAG variants yield MAE values in the range 3.004–3.010, offering only modest gains over Naive-MTL and still clearly underperforming both our method and the OPT oracle.

The proposed method consistently outperforms all other methods with the exception of the Optimal across various task split scenarios. Specifically, in the 4-split scenario, the proposed method achieves the lowest overall MAE.

## C Worked Example: Task Grouping with Constraints

To further elucidate the practical utility of our proposed mathematical programming framework (Formulation 8), we present a concrete, step-by-step worked example. This scenario demonstrates how the MIP formulation effectively resolves task grouping under strict resource constraints, such as GPU memory limitations in a distributed learning setup.

### C.1 Scenario Setup

Consider a multi-task learning scenario involving $n = 6$ tasks, denoted as $\mathcal{T} = \{T_1, \ldots, T_6\}$. These tasks are to be distributed across $m = 3$ GPU nodes ($G_1, G_2, G_3$). Each task entails a specific memory requirement, and each GPU node operates under a distinct memory capacity limit.

**Resource Parameters.** Let $B \in \mathbb{R}^{n \times m}$ represent the budget matrix, where $B_{ij}$ denotes the memory cost of assigning task $i$ to group $j$. In this simplified example, the memory cost is intrinsic to the task and constant across nodes. Let $\mathbf{b} \in \mathbb{R}^m$ represent the capacity vector for the nodes. The specific requirements are detailed below:

Table 8: Grouing and comparison results on the ETTm1 dataset.

| Group | Method | t1 | t2 | t3 | t4 | t5 | t6 | t7 | MAE (↓) |
|---|---|---|---|---|---|---|---|---|---|
| | STL | $0.64 \pm 0.016$ | $0.37 \pm 0.009$ | $0.68 \pm 0.015$ | $0.36 \pm 0.011$ | $0.56 \pm 0.027$ | $0.29 \pm 0.002$ | $0.15 \pm 0.006$ | 3.050 |
| MTL | Bandit-MTL | $0.67 \pm 0.041$ | $0.37 \pm 0.010$ | $0.68 \pm 0.051$ | $0.34 \pm 0.003$ | $0.58 \pm 0.017$ | $0.26 \pm 0.004$ | $0.14 \pm 0.003$ | 3.052 |
| | LS | $0.67 \pm 0.014$ | $0.38 \pm 0.004$ | $0.67 \pm 0.040$ | $0.36 \pm 0.007$ | $0.58 \pm 0.023$ | $0.28 \pm 0.003$ | $0.15 \pm 0.002$ | 3.094 |
| | UW | $0.67 \pm 0.011$ | $0.38 \pm 0.014$ | $0.62 \pm 0.011$ | $0.36 \pm 0.013$ | $0.59 \pm 0.021$ | $0.27 \pm 0.011$ | $0.16 \pm 0.013$ | 3.061 |
| | Nash-MTL | $0.61 \pm 0.017$ | $0.38 \pm 0.011$ | $0.65 \pm 0.006$ | $0.36 \pm 0.004$ | $0.59 \pm 0.037$ | $0.27 \pm 0.003$ | $0.15 \pm 0.013$ | 3.004 |
| | PCGrad | $0.65 \pm 0.033$ | $0.38 \pm 0.011$ | $0.62 \pm 0.020$ | $0.35 \pm 0.001$ | $0.57 \pm 0.006$ | $0.27 \pm 0.011$ | $0.15 \pm 0.001$ | 2.997 |
| | Naive-MTL | $0.66 \pm 0.014$ | $0.38 \pm 0.006$ | $0.64 \pm 0.056$ | $0.35 \pm 0.009$ | $0.59 \pm 0.016$ | $0.28 \pm 0.005$ | $0.15 \pm 0.009$ | 3.038 |
| | Surrogate | $0.65 \pm 0.015$ | $0.38 \pm 0.006$ | $0.64 \pm 0.056$ | $0.35 \pm 0.003$ | $0.59 \pm 0.016$ | $0.28 \pm 0.005$ | $0.15 \pm 0.009$ | 3.040 |
| | STG | $0.65 \pm 0.027$ | $0.38 \pm 0.005$ | $0.66 \pm 0.033$ | $0.36 \pm 0.008$ | $0.59 \pm 0.029$ | $0.27 \pm 0.016$ | $0.16 \pm 0.019$ | 3.065 |
| 2 Groups | Random | $0.66 \pm 0.014$ | $0.38 \pm 0.006$ | $0.64 \pm 0.056$ | $0.35 \pm 0.009$ | $0.59 \pm 0.016$ | $0.28 \pm 0.005$ | $0.15 \pm 0.009$ | 3.035 |
| | OPT | - | $0.37 \pm 0.007$ | - | $0.35 \pm 0.003$ | $0.56 \pm 0.029$ | - | $0.15 \pm 0.011$ | 2.926 |
| | | $0.61 \pm 0.015$ | - | $0.62 \pm 0.036$ | $0.36 \pm 0.007$ | $0.57 \pm 0.036$ | $0.28 \pm 0.008$ | - | |
| | TAG | $0.66 \pm 0.041$ | $0.37 \pm 0.005$ | $0.66 \pm 0.022$ | $0.36 \pm 0.007$ | - | - | - | 3.032 |
| | | $0.66 \pm 0.014$ | $0.38 \pm 0.006$ | $0.64 \pm 0.056$ | $0.35 \pm 0.009$ | $0.59 \pm 0.016$ | $0.28 \pm 0.005$ | $0.15 \pm 0.009$ | |
| | MTG | - | - | - | - | - | - | $0.15 \pm 0.006$ | 3.033 |
| | | $0.66 \pm 0.014$ | $0.38 \pm 0.006$ | $0.64 \pm 0.056$ | $0.35 \pm 0.009$ | $0.59 \pm 0.016$ | $0.28 \pm 0.005$ | $0.15 \pm 0.009$ | |
| | Grad-TAG | $0.65 \pm 0.025$ | - | $0.66 \pm 0.030$ | - | - | $0.28 \pm 0.014$ | - | 3.004 |
| | | - | $0.37 \pm 0.007$ | - | $0.35 \pm 0.003$ | $0.56 \pm 0.029$ | - | $0.15 \pm 0.011$ | |
| | Ours | - | - | $0.65 \pm 0.055$ | - | $0.56 \pm 0.011$ | $0.27 \pm 0.009$ | $0.16 \pm 0.006$ | 2.989 |
| | | $0.61 \pm 0.015$ | $0.38 \pm 0.012$ | - | $0.36 \pm 0.014$ | $0.57 \pm 0.010$ | - | - | |
| 3 Groups | Random | $0.67 \pm 0.025$ | $0.39 \pm 0.004$ | $0.65 \pm 0.031$ | $0.35 \pm 0.004$ | $0.57 \pm 0.020$ | $0.28 \pm 0.009$ | $0.14 \pm 0.001$ | 3.054 |
| | OPT | - | - | - | $0.35 \pm 0.018$ | $0.55 \pm 0.022$ | $0.27 \pm 0.015$ | - | 2.913 |
| | | - | $0.37 \pm 0.007$ | - | $0.35 \pm 0.003$ | $0.56 \pm 0.029$ | - | $0.15 \pm 0.011$ | |
| | | $0.61 \pm 0.015$ | - | $0.62 \pm 0.036$ | $0.36 \pm 0.007$ | $0.57 \pm 0.036$ | $0.28 \pm 0.008$ | - | |
| | TAG | $0.64 \pm 0.016$ | $0.38 \pm 0.008$ | $0.65 \pm 0.017$ | - | - | - | - | 3.037 |
| | | $0.64 \pm 0.026$ | - | $0.67 \pm 0.025$ | $0.36 \pm 0.007$ | $0.58 \pm 0.021$ | - | - | |
| | | $0.63 \pm 0.030$ | - | $0.66 \pm 0.037$ | $0.37 \pm 0.016$ | $0.57 \pm 0.022$ | $0.29 \pm 0.016$ | $0.16 \pm 0.001$ | |
| | MTG | - | - | - | - | - | $0.29 \pm 0.002$ | - | 3.033 |
| | | - | - | - | - | - | - | $0.15 \pm 0.006$ | |
| | | $0.66 \pm 0.014$ | $0.38 \pm 0.006$ | $0.64 \pm 0.056$ | $0.35 \pm 0.009$ | $0.59 \pm 0.016$ | $0.28 \pm 0.005$ | $0.15 \pm 0.009$ | |
| | Grad-TAG | $0.65 \pm 0.025$ | - | $0.66 \pm 0.030$ | - | - | $0.28 \pm 0.014$ | - | 3.005 |
| | | - | $0.38 \pm 0.008$ | - | $0.35 \pm 0.007$ | - | - | $0.14 \pm 0.001$ | |
| | | - | - | - | - | $0.56 \pm 0.027$ | - | - | |
| | Ours | - | - | $0.65 \pm 0.055$ | - | $0.56 \pm 0.011$ | $0.27 \pm 0.009$ | $0.16 \pm 0.006$ | 2.979 |
| | | $0.61 \pm 0.015$ | $0.38 \pm 0.012$ | - | $0.36 \pm 0.014$ | $0.57 \pm 0.010$ | - | - | |
| | | - | $0.37 \pm 0.007$ | - | - | $0.58 \pm 0.037$ | $0.27 \pm 0.012$ | - | |
| 4 Groups | Random | $0.64 \pm 0.026$ | $0.38 \pm 0.005$ | $0.68 \pm 0.043$ | $0.35 \pm 0.005$ | $0.55 \pm 0.022$ | $0.27 \pm 0.005$ | $0.15 \pm 0.011$ | 3.022 |
| | OPT | $0.66 \pm 0.028$ | - | - | - | - | - | $0.14 \pm 0.005$ | 2.906 |
| | | - | - | - | $0.35 \pm 0.018$ | $0.55 \pm 0.022$ | $0.27 \pm 0.015$ | - | |
| | | - | $0.37 \pm 0.007$ | - | $0.35 \pm 0.003$ | $0.56 \pm 0.029$ | - | $0.15 \pm 0.011$ | |
| | | $0.61 \pm 0.015$ | - | $0.62 \pm 0.036$ | $0.36 \pm 0.007$ | $0.57 \pm 0.036$ | $0.28 \pm 0.008$ | - | |
| | TAG | $0.64 \pm 0.016$ | $0.38 \pm 0.008$ | $0.65 \pm 0.017$ | - | - | - | - | 3.026 |
| | | $0.63 \pm 0.004$ | - | $0.63 \pm 0.030$ | - | $0.58 \pm 0.025$ | $0.28 \pm 0.008$ | - | |
| | | - | - | $0.68 \pm 0.039$ | $0.36 \pm 0.013$ | - | - | - | |
| | | - | - | $0.70 \pm 0.054$ | - | - | - | $0.16 \pm 0.007$ | |
| | MTG | $0.64 \pm 0.016$ | - | - | - | - | - | - | 3.018 |
| | | - | - | - | - | - | $0.29 \pm 0.002$ | - | |
| | | - | - | - | - | - | - | $0.15 \pm 0.006$ | |
| | | $0.66 \pm 0.014$ | $0.38 \pm 0.006$ | $0.64 \pm 0.056$ | $0.35 \pm 0.009$ | $0.59 \pm 0.016$ | $0.28 \pm 0.005$ | $0.15 \pm 0.009$ | |
| | Grad-TAG | $0.64 \pm 0.005$ | - | - | - | - | $0.28 \pm 0.011$ | - | 3.010 |
| | | - | $0.38 \pm 0.008$ | - | $0.35 \pm 0.007$ | - | - | $0.14 \pm 0.001$ | |
| | | - | - | $0.68 \pm 0.015$ | - | - | - | - | |
| | | - | - | - | - | $0.56 \pm 0.027$ | - | - | |
| | Ours | - | $0.37 \pm 0.007$ | - | - | $0.58 \pm 0.037$ | $0.27 \pm 0.012$ | - | 2.966 |
| | | - | - | $0.65 \pm 0.055$ | - | $0.56 \pm 0.011$ | $0.27 \pm 0.009$ | $0.16 \pm 0.006$ | |
| | | - | - | - | - | $0.58 \pm 0.016$ | $0.27 \pm 0.008$ | $0.15 \pm 0.005$ | |
| | | $0.61 \pm 0.015$ | $0.38 \pm 0.012$ | - | $0.36 \pm 0.014$ | $0.57 \pm 0.010$ | - | - | |

Table 9: Task Memory Requirements and Node Capacities for the Worked Example

| Task | Memory (GB) | Node | Capacity (GB) |
|------|-------------|------|---------------|
| $T_1$ | 4 | $G_1$ | 10 |
| $T_2$ | 3 | $G_2$ | 10 |
| $T_3$ | 5 | $G_3$ | 8 |
| $T_4$ | 2 | | |
| $T_5$ | 6 | | |
| $T_6$ | 3 | | |

Based on Table 9, the budget matrix $B$ and limit vector $\mathbf{b}$ are formally defined as:

$$B = \begin{bmatrix} 4 & 4 & 4 \\ 3 & 3 & 3 \\ 5 & 5 & 5 \\ 2 & 2 & 2 \\ 6 & 6 & 6 \\ 3 & 3 & 3 \end{bmatrix}, \quad \mathbf{b} = \begin{bmatrix} 10 \\ 10 \\ 8 \end{bmatrix}. \tag{10}$$

**Transfer Gain Matrix.** Assume the cumulative transfer gains $\mathcal{S} \in \mathbb{R}^{6 \times 6}$ have been collected via the procedure described in Section 4.1. For this example, we utilize the following values:

$$\mathcal{S} = \begin{bmatrix} 0.5 & 0.3 & 0.1 & 0.2 & 0.1 & 0.2 \\ 0.3 & 0.6 & 0.2 & 0.1 & 0.0 & 0.3 \\ 0.1 & 0.2 & 0.7 & 0.4 & 0.5 & 0.1 \\ 0.2 & 0.1 & 0.4 & 0.5 & 0.3 & 0.2 \\ 0.1 & 0.0 & 0.5 & 0.3 & 0.8 & 0.1 \\ 0.2 & 0.3 & 0.1 & 0.2 & 0.1 & 0.4 \end{bmatrix}. \tag{11}$$

### C.2 Constraint Verification and Solution

The objective is to maximize the group transfer gain subject to the constraints defined in Formulation 8, specifically the resource constraint $(B \odot X)^\top \mathbf{1} \le \mathbf{b}$. An optimal solution $X^* \in \{0,1\}^{6 \times 3}$ identified for this configuration is:

$$X^* = \begin{bmatrix} 1 & 0 & 0 \\ 1 & 0 & 0 \\ 0 & 1 & 0 \\ 1 & 0 & 0 \\ 0 & 0 & 1 \\ 0 & 1 & 0 \end{bmatrix}. \tag{12}$$

This assignment matrix corresponds to the following task grouping topology:

- **Group 1 ($G_1$):** $\{T_1, T_2, T_4\}$

- **Group 2 ($G_2$):** $\{T_3, T_6\}$

- **Group 3 ($G_3$):** $\{T_5\}$

**Verification of Constraints.** We explicitly verify that $X^*$ satisfies all imposed conditions:

1. **Assignment Completeness ($X\mathbf{1} \ge \mathbf{1}$):** The sum of each row in $X^*$ is exactly 1, ensuring every task is assigned to a group.

2. **Non-Empty Groups ($X^\top \mathbf{1} \geq \mathbf{1}$):** The column sums are $[3, 2, 1]^\top$, confirming that no GPU node is left idle.

3. **Memory Constraints ($(B \odot X)^\top \mathbf{1} \leq \mathbf{b}$):**

   - Node $G_1$: $4(T_1) + 3(T_2) + 2(T_4) = 9 \leq 10$. (Satisfied)
   - Node $G_2$: $5(T_3) + 3(T_6) = 8 \leq 10$. (Satisfied)
   - Node $G_3$: $6(T_5) = 6 \leq 8$. (Satisfied)

4. **Distinctness:** All columns of $X^*$ are distinct, satisfying $\|X_{\cdot j_1} - X_{\cdot j_2}\|^2 \geq 1$.

## C.3 Objective Calculation

The objective function in Formulation 8 calculates the sum of average group transfer gains: $J(X) = \sum_{j=1}^{m} \frac{1}{\mathbf{1}^\top X_{\cdot j}} X_{\cdot j}^\top \mathcal{S} X_{\cdot j}$. We compute the contribution of each group:

- **Group 1 ($|G_1| = 3$):** The subset indices are $\{1, 2, 4\}$.

$$X_{\cdot 1}^\top \mathcal{S} X_{\cdot 1} = \mathcal{S}_{11} + \mathcal{S}_{22} + \mathcal{S}_{44} + 2(\mathcal{S}_{12} + \mathcal{S}_{14} + \mathcal{S}_{24})$$
$$= 0.5 + 0.6 + 0.5 + 2(0.3 + 0.2 + 0.1) = 2.8.$$

  Group contribution: $2.8/3 \approx 0.933$.

- **Group 2 ($|G_2| = 2$):** The subset indices are $\{3, 6\}$.

$$X_{\cdot 2}^\top \mathcal{S} X_{\cdot 2} = \mathcal{S}_{33} + \mathcal{S}_{66} + 2(\mathcal{S}_{36})$$
$$= 0.7 + 0.4 + 2(0.1) = 1.3.$$

  Group contribution: $1.3/2 = 0.65$.

- **Group 3 ($|G_3| = 1$):** The subset index is $\{5\}$.

$$X_{\cdot 3}^\top \mathcal{S} X_{\cdot 3} = \mathcal{S}_{55} = 0.8.$$

  Group contribution: $0.8/1 = 0.8$.

**Total Objective Score:** $0.933 + 0.65 + 0.8 \approx 2.383$.

This worked example illustrates the capability of our framework to identify high-affinity task groups while strictly adhering to heterogeneous resource constraints, a critical feature for scalable deployment in real-world infrastructure.

# D    Discussions on Computational Complexity

In this section, we provide detailed derivations and additional experimental results related to the computational complexity of the proposed method.

## D.1 Derivations

Following the notations in Table 10, we derive the computational complexity as follows: There are four parts of computation to collect the transfer gains for TAG and our method: (1) Loss computation $\{L_j(\phi_{\{j\}}^t, \theta_j^t), \forall j\}$ needs the computation of $n\mathcal{F}$ for both methods; (2) Gradient Computation, $\{\nabla L_j(\phi_{\{j\}}^t, \theta_j^t), \forall j\}$ needs the computation of $n\mathcal{B}$ for both methods; (3) Update parameters: $\{(\phi_{\{j\}}^{t+1}, \theta_j^t), \forall j\}$ for TAG and $\{(\phi_{\{i,j\}}^{t+1}, \theta_j^{t+1}), (\phi_{\{j\}}^{t+1}, \theta_j^{t+1}), \forall i, j\}$ for our method, resulting the computation of $n\mathcal{C}$ and $(\frac{(1+n)n}{2} + n)\mathcal{C}$, respectively; (4) High-order loss computation: $\{L_j(\phi_{\{i\}}^{t+1}, \theta_j^t), \forall i, j\}$ for TAG and $\{L_j(\phi_{\{j\}}^{t+1}, \theta_j^t), L_j(\phi_{\{i,j\}}^{t+1}, \theta_j^{t+1}), \forall i, j\}$ for our method, resulting the computation of $n^2\mathcal{F}$ and $(n + n^2)\mathcal{F}$, respectively. Detailed comparison of computation results are demonstrated in Table 11.

Table 10: Computation complexity of basic operators.

| Task Num | Avg. Dim. Param. | Avg. Complexity of FF | Avg. Complexity of BP |
|---|---|---|---|
| $n$ | $\mathcal{C}$ | $\mathcal{F}$ | $\mathcal{B}$ |

Table 11: Complexity Computation

| Method | Loss Comput. | Grad. Comput. | Update Params. | High-order Loss | In Total |
|---|---|---|---|---|---|
| TAG | $n\mathcal{F}$ | $n\mathcal{B}$ | $n\mathcal{C}$ | $n^2\mathcal{F}$ | $(n^2+n)\mathcal{F}+n\mathcal{B}+n\mathcal{C}$ |
| Ours | $n\mathcal{F}$ | $n\mathcal{B}$ | $\frac{n(n+3)}{2}\mathcal{C}$ | $(n^2+n)\mathcal{F}$ | $(n^2+2n)\mathcal{F}+n\mathcal{B}+\frac{n(n+3)}{2}\mathcal{C}$ |
| Ours (Sampling) | $n\mathcal{F}$ | $n\mathcal{B}$ | $\frac{(n+1)(n+5)}{6}\mathcal{C}$ | $\frac{(n+1)(n+2)}{3}\mathcal{F}$ | $\frac{n^2+6n+2}{3}\mathcal{F}+n\mathcal{B}+\frac{(n+1)(n+5)}{6}\mathcal{C}$ |

## D.2 Complexity of Sampling Strategy

Based on the results in Appendix D.1, although the computational complexity of our approach is on the same order of magnitude as TAG, the time required for transfer gains collection may become a computational bottleneck when the number of tasks is large. To address this issue, we propose a sampling-based method: Define a random variable $T$ that follows a uniform distribution and denote as $T \sim \text{Unif}(\{1, 2, ..., n\})$. During the training, we randomly select a subset of tasks with the size of $T$ and transfer gains are gathered solely from this subset. Then the the computational cost of collecting transfer gains is:

$$\mathbb{E}\left[\frac{T(T+3)}{2}\mathcal{C} + (T^2+T)\mathcal{F}\right] = \frac{(n+1)(n+5)}{6}\mathcal{C} + \frac{(n+1)(n+2)}{3}\mathcal{F},$$

which significantly reduces the computational cost of our method and is substantially lower than that of TAG because $\mathcal{C} \ll \mathcal{F}$ in practice.

# E Distinction from Classic Distance and Covariance-Based Methods

While our proposed formulation for transfer gain shares a high-level objective with existing methodologies—namely, quantifying the relationships between tasks—it possesses fundamental distinctions from classic distance-based and covariance-based methods in terms of measurement mechanism, temporal collection, and theoretical grounding. In this section, we elucidate these differences to clarify the unique position of our approach.

## E.1 Distinction from Distance-Based Methods

Distance-based methods, such as Taskonomy (Zamir et al., 2018) and Representation Similarity Analysis (Dwivedi & Roig, 2019), generally rely on static feature comparisons. Table 12 summarizes the primary divergences between these approaches and our method.

Table 12: Comparison between Distance-Based Methods and Our Transfer Gain.

| Dimension | Distance-Based Methods | Our Transfer Gain |
|---|---|---|
| **Measurement Mechanism** | Static representation similarity | Dynamic training impact |
| **Temporal Collection** | Post-hoc (after pre-training) | Online (during training) |
| **Signal Reflection** | Feature space distance | Actual loss improvement |
| **Theoretical Guarantee** | Heuristic | Proposition 1 |

The fundamental distinctions are threefold:

- **Dynamic vs. Static Nature:** Distance-based methods typically compute fixed task representation similarities (e.g., the cosine similarity of feature embeddings). In contrast, our metric $\mathcal{S}_{i \to j}^t$ is collected

at every training step $t$, thereby capturing the dynamic evolution of task relationships as the model parameters update during the optimization process.

- **Direct Performance Linkage:** Distance methods rely on the indirect hypothesis that tasks with similar feature representations should be grouped together. Our approach establishes a more direct linkage: Observation 1 explicitly guarantees that a higher transfer gain corresponds to a greater actual loss decrease. Furthermore, Proposition 1 provides a theoretical bound on the approximation of group transfer gains, a guarantee absent in heuristic distance measures.

- **Computational Efficiency:** Methodologies like Taskonomy require pre-training $O(n^2)$ independent pairwise transfer learning models to establish relationships. Conversely, our method collects transfer gains during a single joint training run, which is computationally more efficient and more accurately reflects the dynamics of real Multi-Task Learning (MTL) scenarios.

### E.2 Distinction from Covariance and Gradient-Based Methods

Gradient-based methods, such as PCGrad (Yu et al., 2020), Nash-MTL (Navon et al., 2022), and Conflict-Averse Learning (Liu et al., 2021a), focus on modulating the optimization process to mitigate negative transfer. Table 13 highlights the distinctions.

Table 13: Comparison between Covariance/Gradient-Based Methods and Our Transfer Gain.

| Dimension | Covariance/Gradient Methods | Our Transfer Gain |
|---|---|---|
| **Focus** | Gradient conflict resolution | Joint training effectiveness |
| **Application Stage** | Optimization (dynamic adjustment) | Grouping (static planning) |
| **Information Used** | Instantaneous gradients | Cumulative training signals |
| **Objective** | Mitigate negative transfer | Optimal task assignment |

**Key Differences:**

- **Optimization vs. Grouping:** Gradient methods primarily address the question of *how to optimize* efficiently within a given grouping by resolving instantaneous gradient conflicts. Our method addresses the architectural question of *how to determine the optimal grouping* itself.

- **Complementarity:** These approaches are not mutually exclusive but rather complementary. Our method can be employed to construct the optimal task groups, while gradient-based methods like PCGrad can be utilized as the specific optimization strategy within those groups to further enhance performance.

## F Limitations and Future Works

While our method shows promising results, several limitations warrant discussion. First, the scalability of our approach with an extremely large number of tasks remains to be fully explored. That said, two natural extensions to modern large-model settings are particularly promising. **(1) Instruction tuning for LLMs**: different instruction types (e.g., question answering, summarisation, code generation) can be treated as distinct tasks, and transfer gains can be estimated during a lightweight warm-up phase to guide multi-task instruction data scheduling. **(2) Parameter-efficient fine-tuning**: in LoRA- or adapter-based approaches, task-specific adapter parameters naturally serve as $\theta_i$ in our framework while the shared backbone plays the role of $\phi$, enabling principled task grouping with substantially reduced computational overhead. We leave empirical validation of these extensions to future work. Second, although our Mixed Integer Programming (MIP) framework demonstrates versatility, it currently focuses primarily on knapsack constraints for resource allocation. Introducing additional constraint types—such as task dependencies, hierarchical relationships, or temporal constraints—could broaden its applicability and better address complex real-world scenarios. Finally, our transfer gain estimation, while robust, still requires training auxiliary models, which incurs

additional computational overhead during the task grouping phase. Future work could explore more efficient approximation methods or online adaptation strategies to further reduce this cost.

## G   Broader Impact

This research on Multi-Task Learning (MTL) presents a novel approach to task grouping that achieves significant efficiency gains in both academic and industrial settings. It stands out for its flexibility in adapting to diverse and realistic demands, which is crucial for managing complex tasks efficiently. This adaptability is particularly important in the context of growing computational demands in large-scale data analysis. However, the approach also brings forth ethical considerations. The interpretation of inter-task affinities, if not handled cautiously, could lead to incorrect associations or biases, especially in sensitive contexts. It is imperative to recognize and address these risks to prevent potential misuse and ensure the responsible application of this technology. Despite these considerations, the method's ability to considerably reduce computational demands, while catering to specific requirements and maintaining high accuracy, is a noteworthy advancement in MTL.

