# OpenReview forum: "Towards Principled Task Grouping for Multi-Task Learning"
_TMLR — Accepted by TMLR_

### Review · Reviewer_aZ3n · 2026-03-16

**Summary Of Contributions:**

This paper studies Multi-Task Learning and proposes a novel grouping method that utilizes the correlation between tasks in each group. Compared with previous methods, this work proposes a novel concept of 'transfer gains', and the method does not require restrictive assumptions regarding the relationship between tasks.

The authors also present a flexible mathematical programming formulation that accommodates a wide range of resource constraints. In addition, experimental results support the efficiency of the method.

**Audience:**

Yes

**Audience Explanation:**

Yes, at least a portion of TMLR’s audience would be interested in these findings because Multi-Task Learning (MTL) remains a fundamental challenge in balancing efficiency and generalization, particularly when moving away from restrictive, predefined task relationships.

**Claims And Evidence:**

No

**Claims Explanation:**

The experimental results support the efficiency of the method. However, the discussion in the Preliminary and Method sections lacks clear mathematical definitions, which makes it confusing.

First, regarding the preliminaries, the authors provide a global optimization objective (1) and an optimization objective (2). However, objective (1) features shared parameters for all tasks, whereas objective (2) appears to have different shared parameters within each group. Therefore, optimization objective (2) is not directly related to optimization objective (1). Furthermore, with different group divisions, the instances of optimization objective (2) are also unrelated. Under these circumstances, it is difficult to understand the performance of a group optimization result or the group selection method in Section 4.2. Are the authors missing a discussion that translates the results in each group back to the global optimization objective?

Second, Definition 3 for 'Transfer Gain' is highly unclear. In Equation (3), are the $\theta_j^{t+1}$ terms in the numerator and denominator the same? According to the text description, it seems the $\theta_j^{t+1}$ in the numerator starts from $\theta_j^{t}$ and takes a gradient step with the loss from both $T_i$ and $T_j$, while the denominator only considers the gradient step with the loss from $T_j$. However, the notations used are identical. Additionally, there is a missing definition for $\phi$ and $\theta$ in (4) regarding group transfer.

**Requested Changes:**

1. Clarity of Evidence: As discussed in the 'Accurate, Convincing, and Clear' section, the authors should provide more detailed explanations for the methodology and the preliminaries.

2. Assumption in Proposition 1: Regarding the second assumption in Proposition 1, does it require a lower bound specifically for a single step of gradient descent with Task $j$, or must it hold for a single step of gradient descent with any arbitrary group $A$?

3. Optimization in Equation (8): The optimization formulation in (8) requires further explanation. The motivation appears to be the maximization of the total 'group transfer gain.' However, this motivation seems to violate the global optimization objective (1). Specifically, transfer gain is defined as a ratio of losses. For example, if Task $i$ reduces the loss of Task $j$ from 100 to 50, the gain is 0.5. If it reduces the loss from 2 to 1, the gain is still 0.5. In contrast, the global optimization objective (1) considers the summation of losses—a linear relationship. These two cases, while having the same transfer gain, yield totally different improvement in the global objective.
Therefore, the transfer gain metric does not appear to relate directly to the primary optimization target.

4. Empirical Complexity: The authors discuss the theoretical time complexity in Section 4.3. However, it would be beneficial for the authors to provide the actual empirical time complexity observed during experiments and compare it with the actual wall-clock time of baseline methods, such as TAG.

---

> ### Author Response · Authors · 2026-05-09
>
> We sincerely thank Reviewer for the careful and technically detailed reading of our work. We address each point below.
>
> ---
>
> **RC1: Relationship Between Objective (1) and Objective (2)**
>
> We thank the reviewer for raising this point, and agree that the relationship between the two objectives deserves a more explicit treatment.
>
> **The connection between Eq.(1) and Eq.(2).** Eq.(1) is the standard MTL objective where all tasks share a single global parameter $\varphi$. The key motivation for task grouping is that directly optimizing Eq.(1) across all tasks simultaneously often leads to negative transfer — gradients from unrelated tasks interfere with each other, degrading performance across the board. Task grouping addresses this by partitioning tasks into groups such that within each group, tasks exhibit strong positive transfer, while harmful cross-group interactions are eliminated by training each group with its own shared parameter $\varphi_j$.
>
> **Task grouping as a generalization of MTL.** Eq.(2) is a generalization of Eq.(1): when $m=1$, i.e., all tasks are placed in a single group, $\varphi_1 = \varphi$ and Eq.(2) reduces exactly to Eq.(1). Task grouping therefore strictly expands the space of training configurations beyond standard MTL. When tasks exhibit heterogeneous transfer relationships, a well-chosen grouping under Eq.(2) can mitigate negative transfer that would otherwise occur under Eq.(1), leading to better per-task performance. This is empirically supported by Figure 1, where our method consistently outperforms the MTL baseline across all four domains and all split configurations.
>
> **On the apparent unrelatedness of different grouping instances.** The reviewer notes that different group divisions yield different instances of Eq.(2) that appear unrelated. This is precisely the reason why a principled task grouping framework is needed — the goal of Section 4.2 is to search over all possible grouping configurations and identify the one that maximizes the aggregate transfer gain under given constraints. The mathematical programming formulation in Eq.(8) is designed exactly for this purpose.
>
> We will add an explicit paragraph in Section 3 clarifying the relationship between Eq.(1) and Eq.(2), including the generalization argument, to make this connection immediately transparent to readers.
>
> ---
>
> **RC2: Clarification of Notation in Definition 3**
>
> We thank the reviewer for identifying this ambiguity, and agree that the notation in Eq.(3) and Eq.(4) requires more precise exposition.
>
> **Clarification of Eq.(3).** As established in the proof of Proposition 1 in Appendix A, $\theta^{t+1}_j$ is treated as a single fixed quantity throughout — specifically, $\tilde{L}_j(\varphi) = L_j(\varphi, \theta^{t+1}_j)$ with $\theta_j$ fixed at step $t+1$. Therefore, $\theta^{t+1}_j$ in the numerator and denominator of Eq.(3) refers to the **same quantity**. The true source of ambiguity lies solely in the shared parameter $\varphi$, which follows two different update paths:
>
> - **Numerator:** $\varphi^{t+1}_{\{i,j\}}$ is obtained by updating $\varphi^t$ using the **joint gradient** of $T_i$ and $T_j$;
> - **Denominator:** $\varphi^{t+1}_{\{j\}}$ is obtained by updating $\varphi^t$ using **only the gradient** of $T_j$.
>
> Both start from the same $\varphi^t$ at step $t$ but diverge due to different task subsets used for the gradient update. We will revise the notation in Definition 3 to make the subscripts of $\varphi$ more explicit, and add a clarifying sentence explaining that $\theta^{t+1}_j$ is a shared fixed reference point in both terms.
>
> **Clarification of Eq.(4).** We confirm that the group transfer gain in Eq.(4) is defined for $A \subseteq \mathcal{T}$ and $j \notin A$, where $A$ is any subset of tasks excluding $T_j$ itself. We will explicitly state these conditions in the revised version to remove the ambiguity noted by the reviewer.
>
> ---
>
> **RC3: Scope of Assumption 2 in Proposition 1**
>
> We thank the reviewer for this precise technical question. Assumption 2 — $L_j(\varphi^{t+1}_{\{j\}}, \theta^{t+1}_j) \geq C > 0$ — is required to hold **only** for the single-task trained parameters $\varphi^{t+1}_{\{j\}}$, i.e., when $T_j$ is trained alone.
>
> The reason is straightforward: as shown in the proof in Appendix A, this lower bound appears exclusively as the **denominator** in the expressions for $S^t_{i \to j}$ and $S^t_{A \to j}$, both of which are normalized by $\tilde{L}\_j(\phi^{t+1}_{j})$. The loss values under any arbitrary group $A \cup \{j\}$ appear only in the numerator and are not required to satisfy any lower bound. This assumption is mild in practice, as standard regularization techniques (e.g., L2 regularization, dropout) and early stopping naturally prevent the single-task loss from converging to zero during training.
>
> We will revise the statement of Proposition 1 to explicitly clarify that Assumption 2 applies only to the $\{j\}$-trained parameters, eliminating this source of ambiguity.

---

> > ### Author Response · Authors · 2026-05-09
> >
> > **RC4: Alignment Between Transfer Gain (Ratio) and Global Objective (Sum)**
> >
> > We thank the reviewer for this insightful observation, which touches on a core design choice of our method. We address this carefully in two layers.
> >
> > **Layer 1 — Within-task ordering consistency.** For a fixed target task $T_j$, comparing two candidate source tasks $T_i$ and $T_k$, the condition $S^t_{i \to j} > S^t_{k \to j}$ is equivalent to:
> >
> > $$\frac{-\eta\_t \nabla\tilde{L}\_j(\xi\_i)^T G^t\_i}{\tilde{L}\_j(\varphi^{t+1}\_{\lbrace j \rbrace})} > \frac{-\eta\_t \nabla\tilde{L}\_j(\xi\_k)^T G^t\_k}{\tilde{L}\_j(\varphi^{t+1}\_{\lbrace j \rbrace})}$$
> >
> > Since the denominator $\tilde{L}\_j(\varphi^{t+1}_{\lbrace j \rbrace})$ is **identical** for both sides when the target task $T_j$ is fixed, it cancels, and the inequality reduces directly to a comparison of **absolute loss decreases** for $T_j$:
> >
> > $$-\eta_t \nabla\tilde{L}_j(\xi_i)^T G^t_i > -\eta_t \nabla\tilde{L}_j(\xi_k)^T G^t_k$$
> >
> > This shows that within a fixed target task, a higher transfer gain correctly identifies the source task that produces a greater absolute loss decrease — which is consistent with the spirit of Observation 1, extended here to the pairwise source task comparison level from the underlying Mean Value Theorem derivation in Appendix A.
> >
> > **Layer 2 — Cross-task aggregation and the role of normalization.** We acknowledge that the denominator cancellation argument above is valid only **within a fixed target task $T_j$**, and does not directly extend to the aggregation across different target tasks in Eq.(8). When summing transfer gains across tasks with different loss scales — for instance, the objective value function in COP and regression loss in Taskonomy operate on entirely different magnitudes — the denominators $\tilde{L}\_j(\varphi^{t+1}_{\lbrace j \rbrace})$ differ across tasks and cannot be jointly cancelled. Therefore, we do not claim that Eq.(8) is strictly equivalent to optimizing the global loss sum in Eq.(1).
> >
> > Rather, Eq.(8) optimizes a **normalized surrogate** that serves as a principled proxy for the global objective. The normalization is a deliberate and necessary design choice: without it, tasks with larger loss magnitudes would systematically dominate the grouping decisions, introducing a scale-dependent bias that would render the affinity measure incomparable across tasks. By normalizing each transfer gain by the corresponding single-task loss $\tilde{L}\_j(\varphi^{t+1}_{\lbrace j \rbrace})$, our formulation ensures that the contribution of each task to the grouping decision is **scale-invariant and comparably meaningful**, regardless of individual loss magnitudes. This normalization design is consistent with prior work such as TAG and HOA, and is empirically validated by the consistent improvements our method achieves across domains with fundamentally different loss scales — including vision, combinatorial optimization, and time-series forecasting.
> >
> > ---
> >
> > **RC5: Empirical Wall-Clock Time Comparison**
> >
> > We thank the reviewer for this suggestion. We note that two pieces of empirical timing evidence are already present in the paper. First, Table 2 provides a direct wall-clock time comparison between our MIQP solver (MP) and Branch & Bound (BB) across all datasets and split configurations. Second, Figure 3 presents a GPU hours vs. performance plot on CelebA that directly compares our method (both full and sampling-enhanced variants) against TAG, CS, and several MTL baselines, showing that our method achieves superior performance within comparable or lower computational budgets.
> >
> > To further address the reviewer's concern, we provide here the actual wall-clock times for transfer gain collection measured during our experiments on CelebA and ETTm1:
> >
> > | Dataset | Method | Total Epochs | Total Wall-Clock Time |
> > |---------|--------|-------------:|----------------------:|
> > | CelebA  | Ours   | 100          | 23.66 min             |
> > | CelebA  | TAG    | 100          | 12.68 min             |
> > | ETTm1   | Ours   | 10           | 175.01 min            |
> > | ETTm1   | TAG    | 10           | 188.17 min            |
> >
> > The results show that the additional computational overhead of our method is modest. On CelebA, our method incurs approximately 1.87× the wall-clock time of TAG during transfer gain collection, which is consistent with the theoretical complexity analysis in Section 4.3 — our method requires additional parameter operations that account for this difference. On ETTm1, our method is actually slightly faster than TAG in absolute terms. Importantly, as demonstrated in Table 1, applying the lazy collection strategy with collection intervals of 10–50 steps achieves 10–15× speedup while maintaining or even improving performance, which would bring the wall-clock time of our method well below that of TAG in both settings. We will incorporate this timing comparison into Section 4.3 or Appendix D in the revised version.

---

### Review · Reviewer_1VUu · 2026-04-20

**Summary Of Contributions:**

The paper introduces a principled framework for task grouping in multi-task learning (MTL) that addresses longstanding limitations in prior work (e.g., TAG and HOA). The core contributions are: (1) a theoretically grounded, assumption-free transfer-gain definition (Eq. 3–5) that measures the effect of joint training without requiring convexity or smoothness, supported by a rigorous bound in Proposition 1 showing that group transfer gain is well-approximated by the average of pairwise gains under standard optimization conditions; (2) a flexible Mixed-Integer Quadratic Programming (MIQP) formulation (Eq. 8–9) solvable by off-the-shelf solvers such as Gurobi, which naturally incorporates arbitrary budget and group-size constraints; (3) two practical efficiency enhancements—sampling and lazy collection—that reduce computational cost by 10–15× while preserving performance; and (4) extensive experiments across computer vision (Taskonomy, CelebA), combinatorial optimization (TSP/CVRP/OP at two scales), and time-series forecasting (ETTm1), consistently outperforming strong baselines including TAG, HOA, MTG, STG, Grad-TAG, and various MTL loss-balancing methods.

**Audience:**

Yes

**Audience Explanation:**

Task grouping remains a central challenge in MTL, especially as models scale to dozens or hundreds of tasks in vision, NLP, robotics, and scientific ML. The community is actively seeking methods that (i) avoid restrictive theoretical assumptions, (ii) handle realistic resource constraints, and (iii) scale beyond small task sets. This work directly addresses all three points with a clean theory-to-practice pipeline, making it highly relevant to TMLR readers working on multi-task architectures, efficient training, and automated transfer learning. The multi-domain validation (CV + COP + time series) further broadens its appeal.

**Broader Impact Concerns:**

None. The work promotes more efficient and resource-aware multi-task learning, which has clear positive societal implications (reduced carbon footprint, better performance on edge devices, and more equitable access to MTL in resource-constrained settings). No ethical risks or dual-use concerns are apparent, and the authors appropriately discuss limitations in Section F. No Broader Impact Statement is required.

**Claims And Evidence:**

Yes

**Claims Explanation:**

The theoretical claims (assumption-free transfer gain and the bound in Proposition 1) are formally proven in Appendix A using only Lipschitz continuity and a mild positive lower bound on the loss—standard and realistic conditions. All experimental claims are backed by comprehensive tables (Tables 3–5 and additional appendix tables), figures (Figs. 1–5), and ablations that isolate the contribution of the new transfer gain (“Ours-MP” vs. “TAG-MP”) and the MIQP solver (“Ours-MP” vs. “Ours-BB”). Results are averaged over multiple random seeds, include recent strong baselines (STG, Grad-TAG, Linear Surrogate), and demonstrate consistent gains across three distinct domains and under explicit resource constraints. The efficiency claims are quantified with concrete speedup numbers and wall-clock comparisons. No cherry-picking is evident; the authors transparently discuss cases where performance plateaus or where certain baselines are competitive.

**Requested Changes:**

All changes are strengthening (none are critical to acceptance).

In Section 4.1, briefly clarify how the metric $  L_j  $ is instantiated in practice for regression vs. classification tasks (e.g., MSE vs. cross-entropy) and confirm that the same choice is used consistently across all domains—this removes any residual ambiguity for readers reimplementing the method.


Minor typographical/notation polish: standardize the notation for group transfer gain $  S_{A\to j}  $ throughout (it is occasionally written with a superscript $  t  $).

These revisions are minor and can be completed quickly; the paper is already in excellent shape.

---

> ### Author Response · Authors · 2026-05-09
>
> We sincerely thank Reviewer 1VUu for the thorough and positive assessment of our work, and for the careful reading reflected in these suggestions. We address both requested changes below.
>
> ---
>
> **RC1: Clarification of How the Metric $L_j$ is Instantiated Across Domains**
>
> We thank the reviewer for raising this point. To remove any ambiguity for readers reimplementing the method, we clarify that $L_j$ is instantiated consistently across all experiments, with the specific form determined by the task type in each domain:
>
> | Dataset | Task Type | $L_j$ |
> |---------|-----------|--------|
> | Taskonomy | Visual regression | Task-specific regression training loss |
> | CelebA | Multi-label classification | Binary cross-entropy training loss |
> | COP | Combinatorial optimization | Objective value function (POMO framework) |
> | ETTm1 | Time-series forecasting | MAE training loss |
>
> We acknowledge that the original phrasing in Definition 3 — "such as the loss function or validation accuracy" — does not make sufficiently clear which instantiation is used in each setting. As shown in the table above, the choice of $L_j$ is determined by what best reflects task performance in each domain: training loss is used where it directly corresponds to the optimization objective, while the objective value function is used for COP where it serves as the natural task-specific signal within the POMO reinforcement learning framework. In the revised version, we will update Definition 3 to explicitly acknowledge this domain-dependent instantiation, and will add a corresponding clarification in Appendix B.2 so that the specific form used in each setting is immediately accessible to readers.
>
> ---
>
> **RC2: Typographical and Notation Polish**
>
> We thank the reviewer for catching this inconsistency. We will standardize the notation for group transfer gain to $S^t_{A \to j}$ throughout the entire manuscript, and will conduct a full pass to ensure all related symbols are used consistently.

---

### Review · Reviewer_RgqP · 2026-05-03

**Summary Of Contributions:**

This paper studies task grouping in multi-task learning. The authors propose a new definition of transfer gain that does not rely on strong assumptions (e.g., convexity), and use it to construct a task affinity matrix. Based on this, they formulate task grouping as a mathematical programming problem (MIQP) that can incorporate various constraints such as budget or group size. Experiments on vision, combinatorial optimization, and time series datasets show improvements over prior grouping methods like TAG.

**Audience:**

Yes

**Audience Explanation:**

There is still a line of work in multitask learning that focuses on task relationships, negative transfer, and grouping. For researchers working specifically in this area, the paper provides a cleaner formulation of transfer gain and a more flexible optimization-based grouping framework, which could be of interest.

However, I do think the impact may be somewhat limited outside this niche. In particular, the connection to modern large-scale multi-task or foundation model settings (e.g., instruction tuning, large task mixtures) is not discussed, which makes it less clear how broadly useful the method is today.

**Broader Impact Concerns:**

I do not see major ethical concerns specific to this work. The paper focuses on improving optimization and task grouping in multitask learning, and does not introduce new risks beyond standard machine learning concerns.

**Claims And Evidence:**

Yes

**Claims Explanation:**

The empirical results are generally consistent with the claims. The proposed method is compared against a wide range of baselines, including MTL methods and prior grouping approaches (e.g., TAG), and shows improvements across multiple datasets. The ablation studies (transfer gain vs. TAG, MP vs. branch-and-bound) are also helpful in isolating where the gains come from.

That said, the evaluation is limited to relatively small-scale and somewhat standard benchmarks (e.g., Taskonomy, CelebA), and mostly follows prior setups. While this is sufficient to support the claims within this experimental scope, it does not fully establish how the method would behave in more realistic or large-scale scenarios.

**Requested Changes:**

- The paper should more clearly articulate what is fundamentally new compared to prior work such as TAG. Right now, the proposed transfer gain appears to be a variant of existing lookahead-style affinity measures, and the main difference is the removal of certain assumptions. It would help to more explicitly discuss whether this leads to qualitatively different behavior, or primarily cleaner theoretical justification.
- The practical relevance needs to be better justified. The current experiments are conducted on small-scale benchmarks with a limited number of tasks. It is unclear how the proposed method scales with the number of tasks or model size, especially given the $O(n^2)$ cost of transfer gain estimation and the use of mixed-integer programming. A discussion (or even rough analysis) of scalability to larger settings would strengthen the paper.

- It would be helpful to include experiments or discussion in more modern settings, or at least explain how the method could be adapted to them. For example, how would this approach apply to large-scale multi-task training or foundation models where tasks are not cleanly separated?

---

> ### Author Response · Authors · 2026-05-09
>
> We sincerely thank Reviewer RgqP for the thoughtful and constructive feedback. We are glad that the reviewer finds the paper's contributions relevant and the empirical results consistent with our claims. Below we address each requested change in detail.
>
> ---
>
> **RC1: More Explicit Discussion of What is Fundamentally New Compared to TAG**
>
> We appreciate this comment and agree that the distinction deserves a more prominent treatment. We argue that our method differs from TAG not merely in offering cleaner theoretical justification, but in ways that lead to qualitatively different empirical behavior.
>
> **Difference in measurement target.** While the two formulations appear superficially similar, they measure fundamentally different quantities. TAG's affinity $Z^t\_{i \to j} = 1 - \frac{L\_j(\varphi^{t+1}\_{\lbrace i \rbrace}, \theta^t\_j)}{L\_j(\varphi^t, \theta^t\_j)}$ measures the effect of training $T\_i$ alone on the evaluation of $T\_j$ at a fixed snapshot $\theta^t\_j$. By contrast, our transfer gain $S^t\_{i \to j} = 1 - \frac{L\_j(\varphi^{t+1}\_{\lbrace i,j \rbrace}, \theta^{t+1}\_j)}{L\_j(\varphi^{t+1}\_{\lbrace j \rbrace}, \theta^{t+1}\_j)}$ measures the effect of jointly training $\lbrace T\_i, T\_j \rbrace$ on the training outcome of $T\_j$, with task-specific parameters also updated. This distinction is not cosmetic: TAG computes a counterfactual quantity (applying $T\_i$'s gradient to evaluate $T\_j$'s fixed parameters), whereas ours directly reflects the actual benefit of co-training in the MTL sense.
>
> **Difference in theoretical guarantees for group-level gains.** TAG defines the group transfer gain as a direct arithmetic average of pairwise affinities $Z^t\_{\lbrace j,k \rbrace \to i} = \frac{1}{2}(Z^t\_{j \to i} + Z^t\_{k \to i})$, without any theoretical justification for why this approximation is accurate. Our Proposition 1 provides an explicit error bound showing that the group transfer gain is well-approximated by the average of individual gains, with the bound controlled by the learning rate, group size, and Lipschitz constant — all of which are small under standard training conditions. This guarantee is what legitimizes the quadratic objective in Formulation 8, something TAG's framework cannot provide.
>
> **Qualitatively different empirical behavior.** Most importantly, the distinction is not merely theoretical — it produces observable performance differences in practice. As shown in Figure 1 and Tables 5–6, TAG's performance on COP and ETTm1 is comparable to or even worse than the random baseline, while our method consistently outperforms all non-optimal baselines on these datasets. The root cause is precisely the assumption gap: TAG requires strong convexity and smoothness of loss functions to establish the ordering property of its affinity measure, but these conditions are unlikely to hold for the combinatorial optimization losses and time-series forecasting losses used in COP and ETTm1. Our transfer gain, by contrast, requires no such conditions (Observation 1 holds unconditionally), which explains why our method remains robust across all four domains while TAG degrades on non-vision tasks.
>
> **Planned revision.** In response to this comment, we will add a dedicated paragraph in Section 4.1 and a concise summary table contrasting our transfer gain with TAG's along three dimensions — measurement target, theoretical guarantee for group-level gains, and behavioral robustness — to make these distinctions immediately clear to readers.

---

> > ### Author Response · Authors · 2026-05-09
> >
> > **RC2: Scalability with Number of Tasks and Use of Mixed-Integer Programming**
> >
> > We appreciate this concern and address the two potential bottlenecks — transfer gain collection and MIQP solving — separately.
> >
> > **Scalability of transfer gain collection.** We have already proposed and empirically validated two efficiency-enhancing strategies in Sections 4.3 and 5.4. The sampling strategy reduces the expected complexity from $O(n^2)$ to $O(n^2/3)$ in terms of feed-forward passes, and the lazy collection strategy achieves 10–15× wall-clock speedup with intervals of 10–50 steps, as shown in Table 1. Notably, on CelebA, reducing collection frequency to every 50 steps actually improves relative performance by 5.13% over random grouping, compared to 3.50% with full collection — suggesting that sparser collection reduces estimation noise. It is also important to note that transfer gain collection is a one-time preprocessing step: once the affinity matrix is obtained, all subsequent group-wise training runs at standard MTL cost.
> >
> > **Scalability of the MIQP solver.** Table 2 already demonstrates that our MIQP formulation (labeled MP) solves all tested configurations within 30 seconds, including 9-task CelebA with up to 6 splits — scenarios where Branch & Bound fails to terminate within 8 hours. For even larger task sets, we note that MIQP problems of comparable or greater complexity are routinely solved in practice by modern solvers. A well-known example is portfolio optimization in finance: the classical Markowitz mean-variance model with cardinality constraints (selecting at most $k$ assets from a universe) is structurally a MIQP with binary variables, and practitioners regularly solve instances with hundreds of assets using Gurobi as a standard workflow. Our task grouping MIQP is considerably smaller in scale than such problems. Furthermore, our formulation inherits the special structure of an assignment problem — each task is assigned to exactly one group — for which modern solvers have dedicated branching and cutting-plane strategies that substantially accelerate convergence beyond general MIQP. For settings with very large task counts (e.g., $n > 50$), LP relaxation with rounding or hierarchical grouping strategies are natural extensions, which we have noted as future work in Section F.
> >
> > ---
> >
> > **RC3: Applicability to Modern Large-Scale Settings such as Foundation Models**
> >
> > We acknowledge that the connection to modern large-scale multi-task or foundation model settings is not discussed in the paper, and we appreciate the reviewer raising this point.
> >
> > We first note that our experimental scope — 5 to 9 clearly delineated tasks across four domains — is consistent with the standard benchmarking protocol of prior task grouping work (TAG, HOA, MTG), which is necessary for controlled, comparable evaluation. That said, we agree that discussing how the method could extend to larger settings adds important context for TMLR readers.
> >
> > We see two natural pathways for extending our framework to modern settings. First, in instruction tuning of large language models, different instruction types (e.g., mathematical reasoning, code generation, dialogue) can be treated as distinct tasks, each with a dedicated task-specific training loss or metric. Our transfer gain can be estimated during a short warm-up phase on a representative data mixture, and the resulting groupings can guide data scheduling or staged training, without requiring architectural changes. Second, in parameter-efficient fine-tuning (e.g., LoRA or adapter-based methods), task-specific adapter parameters naturally serve as $\theta\_i$ in our formulation, while the shared backbone acts as $\varphi$. In this case, transfer gain collection requires only adapter-level parameter operations, substantially reducing the computational overhead compared to full fine-tuning.
> >
> > We will add a paragraph in Conclusions or Section F (Limitations and Future Work) to outline these potential extensions, making the broader applicability of our framework more explicit for readers working in large-model settings.

---

### Decision · Action_Editor_ENiy · 2026-06-08

**Recommendation:** Accept with minor revision

**Additional Comments:**

The paper is technically solid, well-executed, and makes a meaningful contribution to the task grouping literature by removing restrictive assumptions present in prior work like TAG while providing both theoretical guarantees and practical scalability considerations. The authors’ rebuttal was thorough and responsive, particularly in clarifying distinctions from TAG (measurement target, theoretical bounds, and empirical robustness), addressing scalability, and outlining extensions to large-scale/foundation model settings.

Minor revisions requested (should be straightforward to address):

Incorporate the planned clarifications and comparison table contrasting the proposed transfer gain with TAG (as described in the authors’ response to RgqP).

Add the table/instantiation details for the performance metric across domains (regression loss, cross-entropy, objective value, MAE) in Definition 3 and Appendix B.2.

Standardize notation for group transfer gain as discussed.

Include the brief discussion on extensions to instruction tuning / LoRA-style settings in the Conclusions or Limitations section.

Optionally add the empirical wall-clock time comparisons for transfer gain collection (already provided in the rebuttal) to strengthen the efficiency claims.

These changes will further improve clarity and positioning without requiring new experiments.

**Audience:**

Yes

**Audience Explanation:**

Task grouping and mitigating negative transfer remain important challenges in MTL, particularly as models scale to larger task collections in vision, NLP, robotics, and scientific applications. This work offers a clean theoretical framework, a flexible optimization-based grouping method that handles realistic constraints, and practical efficiency improvements. It will be of clear interest to researchers working on multi-task architectures, transfer learning, efficient training, and automated task relationship modeling. The multi-domain results and discussions on extensions to modern settings (e.g., instruction tuning and parameter-efficient fine-tuning) further broaden its appeal.

**Claims And Evidence:**

Yes

**Claims Explanation:**

The paper presents a principled, assumption-free definition of transfer gain for task grouping in multi-task learning (MTL), supported by a clear theoretical analysis. The MIQP formulation for constrained task grouping is technically sound and solvable with off-the-shelf solvers. Empirical validation is strong and multi-domain (vision, combinatorial optimization, time series), with comprehensive comparisons against strong baselines, ablation studies isolating the contributions of the new transfer gain and the MIQP solver, and efficiency enhancements that deliver 10–15× speedups. Authors have adequately addressed reviewer concerns regarding clarity of notation, relationship between objectives, and distinctions from TAG in the rebuttal and planned revisions. While experiments are on moderate-scale benchmarks, the claims are well-supported within the evaluated scope.